# A Quotient Homology Theory of Representation in Neural Networks

## Abstract

Previous research has proven that the set of maps implemented by neural networks with a ReLU activation function is identical to the set of piecewise linear continuous maps. Furthermore, such networks induce a hyperplane arrangement splitting the input domain of the network into convex polyhedra $G_J$ over which a network $\Phi$ operates in an affine manner.

In this work, we leverage these properties to define an equivalence class $\sim_\Phi$ on top of an input dataset, which can be split into two sets related to the local rank of $\Phi_J$ and the intersections $\cap \mathrm{Im} \Phi_{J_i}$. We refer to the latter as the *overlap decomposition* $\mathcal{O}_\Phi$ and prove that if the intersections between each polyhedron and an input manifold are convex, the homology groups of neural representations are isomorphic to quotient homology groups $H_k(\Phi(\mathcal{M})) \simeq H_k(\mathcal{M}/\mathcal{O}_\Phi)$. This lets us intrinsically calculate the Betti numbers of neural representations without the choice of an external metric. We develop methods to numerically compute the overlap decomposition through linear programming and a union-find algorithm.

Using this framework, we perform several experiments on toy datasets showing that, compared to standard persistent homology, our overlap homology-based computation of Betti numbers tracks purely topological rather than geometric features. Finally, we study the evolution of the overlap decomposition during training on several classification problems while varying network width and depth and discuss some shortcomings of our method.

## 1 Introduction

Deep learning is an incredibly powerful framework for learning and generalizing to highly complex tasks. Despite its widespread success, our theoretical understanding of this framework is still lacking. What we do know is that deep neural networks are universal approximators (Hornik (1991)) and therefore have unbounded expressive power. If we further restrict ourselves to neural networks with ReLU activation functions, we also know that they are exactly equal to the set of continuous piecewise-linear (CPWL) functions (Arora et al. (2018)).

A CPWL function applies a different affine function over different pieces of its domain. Naturally, this phenomenon is also observed in neural networks, where the structure of their weights splits the domain into different polyhedra called *linear regions* (Pascanu et al. (2013); Montufar et al. (2014)). Quantifying the number (Arora et al. (2018); Raghu et al. (2017); Serra et al. (2018); Hanin and Rolnick (2019)) as well as other properties (Fan et al. (2023)) of the linear regions that a network generates is thought to reflect the expressivity of a neural network and is a rich subfield in its own right.

Another avenue of research that has shown promise in many fields of science is Topological Data Analysis (TDA), which attempts to quantify the topological features of data (Carlsson (2009); Wasserman (2018)). In the context of deep learning, it has been used to relate the topology of training trajectories (Birdal et al. (2021); Dupuis et al. (2023)) and network weights (Rieck et al. (2018); Gabrielsson and Carlsson (2019); Gutiérrez-Fandiño et al. (2021); Andreeva et al. (2023)) to the generalization error of a neural network. In addition, while counting linear regions is one measure of expressivity, another way to quantify it is to look at the topology of the decision boundary of a neural network (Guss and Salakhutdinov (2018); Petri and Leitão (2020); Grigsby and Lindsey (2022))

or the degree to which a network can change the topology of the input domain as it is propagated through its layers (Naitzat et al. (2020); Wheeler et al. (2021)).

TDA includes a plethora of methods, among which are persistence landscapes (Bubenik et al. (2015)), persistence images (Adams et al. (2017)), persistent laplacians (Mémoli et al. (2022)) and the Mapper algorithm (Singh et al. (2007)). These and most methods in TDA are based on the concept of persistence and more specifically - the persistent homology method (Zomorodian and Carlsson (2004)). Despite the power of this method, it is known that it is sensitive to outliers, undersampling, and highly non-linear mappings. Furthermore, it is based on calculating distances between points and therefore requires the choice of a metric. This choice leads to the identification of not only topological features, but also geometric ones like curvature (Bubenik et al. (2020)) and convexity (Turkes et al. (2022)). That might be a benefit in some cases, but when reasoning solely about topological features, we would like to avoid contamination from geometric sources.

These issues are especially relevant in the context of deep neural networks as they apply highly non-linear transformations of the input domain that increase curvature (Poole et al. (2016)) in the output space, making their analysis incredibly difficult. To avoid this, we study topological features in the input space through a purely topological lens. We do this using arguments from quotient and relative homology (Hatcher (2002)), which, except for a few works (Pokorny et al. (2016); Blaser and Brun (2022); Beshkov and Einevoll (2024)), have received surprisingly little attention in the deep learning literature.

### 1.1 OUR CONTRIBUTIONS

Instead of dealing with an external space induced by a map, in which the choice of a metric is ambiguous, we take the following approach. Start with a dataset, determine which pieces are glued to each other by the map, and then count the holes in the manifold after this gluing. Under some simple assumptions, these three steps can be rigorously identified with a manifold $\mathcal{M}$, an equivalence class $\sim_\Phi$ and the quotient homology groups $H_k(\mathcal{M}/\sim_\Phi)$ respectively (see Appendix A for background on these terms). This procedure avoids mixing geometric information into our estimates of homology since it only depends on the quotient space $\sim_\Phi$. However, it requires knowledge or at least a good estimate of the topology of the manifold $\mathcal{M}$ and a reliable way to identify $\sim_\Phi$.

In this work, we show how these steps can be carried out in the context of ReLU neural networks, further developing previous work relating polyhedral decompositions and homology (Liu et al. (2023)). There are two difficulties with making such an approach rigorous. Firstly, how can we actually calculate the equivalence class $\sim_\Phi$? Secondly, given this class, how and when can we calculate the quotient homology groups $H_k(\mathcal{M}/\sim_\Phi)$? Our fundamental realization is that since ReLU neural networks split the input domain into convex polyhedra, we can separate the sources of gluing (non-injectivity) into the local rank of the map at each polyhedron and the non-trivial intersections between the images of different polyhedra. We refer to these as the rank and the overlap source, respectively, and show that the latter can be computed exactly, using feasibility checks solved through linear programming. Afterwards, we prove under which conditions the overlap decomposition is sufficient to calculate quotient homology groups.

We compare the performance of persistent and quotient homology when it comes to determining the Betti numbers on toy datasets with a known topology. Our approach gives a new perspective on how topological information propagates through the layers of a neural network. Using quotient homology, we observe that topological simplification happens much more gradually than observed by previous persistent homology-based calculations (Naitzat et al. (2020)). Finally, we study the properties of the overlap decomposition in randomly initialized and trained neural networks and show that the volume of overlapping regions decreases after training, whereas the number of overlap regions tends to increase.

## 2 DECOMPOSITIONS OF NEURAL NETWORKS

We begin by fixing our formalism. We will assume that a dataset $\mathcal{D} = \{x_1, x_2, ..., x_K\}$ is sampled from some compact manifold $\mathcal{M}$, with or without boundary, embedded in $\mathbb{R}^{n_0}$. A neural network is a composite function $\Phi : \mathbb{R}^{n_0} \to \mathbb{R}^{n_1} \to ... \to \mathbb{R}^{n_L}$, where $n_l$ is the $l$-th layer of the network. Between each pair of layers, we have *preactivations* $T_l : \mathbb{R}^{n_{l-1}} \to \mathbb{R}^{n_l}$ given by the affine functions

$T_l(z) = Wz + b$ and *representations* given by $S_l(z) = \text{ReLU}(T_l(z))$. We will denote the output at layer $l$ by $\Phi^l : \mathbb{R}^{n_0} \to \mathbb{R}^{n_l}$ which is given by function composition $\Phi^l = S_l \circ S_{l-1} \circ ... \circ S_1 = \bigcirc_{k=1}^{l} S_k$. We will use the convention that the last layer does not apply an activation function so $\Phi^L = T_L \circ \bigcirc_{k=1}^{L-1} S_k$. Here we use the $\bigcirc$ symbol to denote function composition following the indexing order.

## 2.1 ReLU NETWORKS AND LINEAR REGIONS

As shown in Arora et al. (2018), every ReLU neural network is equivalent to a continuous piecewise-linear function and vice versa. Such a network decomposes input space into convex polyhedra (Montufar et al. (2014); Balestriero et al. (2019); Zhang and Wu (2020); Liu et al. (2023)). This happens as a result of the fact that the affine functions $T_l$ can be interpreted as specifying conditions $CT_l$, where $C$ is some diagonal matrix with entries in $\{-1, 1\}$ defining a polyhedron $P = \{x | CT_l(x) \leq 0\}$. Each subsequent layer further decomposes the input space into more polyhedra, since it adds more conditions to the previous decomposition.

It is also known that within these convex polyhedra, the neural network applies an affine map. For any input $x_k$ in a dataset, we can look at the neurons that get activated by it and write down the *codeword vector* $c_l(x) = \text{sign}(\Phi^l(x))$ and the *precodeword vector* $p_l(x) = \text{sign}(T_l \circ \Phi^{l-1}(x))$ at a layer $l$. Since this is defined at a single layer, we will call it a *local layer codeword*. There are many inputs that activate the same neurons within a layer, so a single codeword corresponds to a whole region of input space. This leads to our first decomposition.

**Definition 2.1.** For a codeword $J \in \{0, 1\}^{n_l}$ and a supporting set $R$ of input space under consideration (for example the dataset $\mathcal{D}$), there is an associated codeword set supported on $R$,

$$L_J^l|_R = \{x | c_l(x) = J, x \in R \subset \mathbb{R}^{n_0}\}. \tag{1}$$

Then the *local layer decomposition* supported on $R$ is defined by the set $\mathcal{L}^l|_R = \{L_J^l|_R | \forall J\}$.

While the codewords within the same region of the local layer decomposition are constant, that does not mean that the network applies the same affine function over the region (see C.1 in the Appendix). However, this does hold in the polyhedral decomposition (Liu et al. (2023)). Instead of looking at the codeword at layer $l$ we can stack all codeword vectors from the previous layers $C_l(x) = [c_1(x), ..., c_l(x)]$ forming a new *global codeword vector*. This leads us to the next decomposition,

**Definition 2.2.** For a codeword $J \in \{0, 1\}^{\sum_{k=1}^{l} n_k}$ and a supporting set $R$ of input space under consideration, there is an associated codeword set supported on $R$,

$$G_J^l|_R = \{x | C_l(x) = J, x \in R \subset \mathbb{R}^{n_0}\}. \tag{2}$$

Then the *polyhedral decomposition* supported on $R$ is defined by the set going over all possible $J$'s or $\mathcal{G}^l|_R = \{G_J^l|_R | \forall J\}$.

As already mentioned, on each polyhedron $G_J^l$ the neural network applies an affine map $\Phi^l|_{G_J^l} : G_J^l \to \mathbb{R}^{n_l}$, which can be written explicitly as,

$$\Phi^l|_{G_J^l}(\cdot) = (\prod_{k=1}^{l} Q_{J_k} W_k)(\cdot) + \sum_{i=1}^{l} \prod_{j=i+1}^{l} (Q_{J_j} W_j) Q_{J_i} b_i, \tag{3}$$

where $Q_{J_k} = \text{diag}(J_k)$ and $J_k$ is given by the codeword on the polyhedra. From now on we shall simplify our notation by writing this map as $\Phi_J^l$.

## 2.2 THE OVERLAP DECOMPOSITION

The polyhedral decomposition is known in the literature. Here we define a new type of decomposition, which we will call the *overlap decomposition*, that can be leveraged to compute the homology groups of a neural network. Our approach is intuitively justified by the realization that if neural networks are equivalent to continuous piecewise-linear functions, then the only way that the topology of the input space can change is through non-injective transformations. Since the network operates piecewise, we can have non-injectivity from two sources; see Appendix B.2 for a proof:

1. The map $\Phi_J^l$ is low-rank and projects the region $G_J^l$ to a lower-dimensional subspace. We call this the *rank source*.

2. For a set of maps $\{\Phi_{J_1}^l, ..., \Phi_{J_N}^l\}$ on different polyhedra, there is a non-trivial intersection $\bigcap_n \text{Im}\Phi_{J_n}^l \neq \emptyset$. We call this the *overlap source*.

The first condition has been described in the work of Beshkov and Einevoll (2024), but as we will see later, it can be ignored when the intersections between the data manifold and the regions $G_J^l$ are convex. Therefore, we will focus on defining the overlap decomposition based on the second condition, with the understanding that one should proceed with caution when this condition fails to hold. Given this, let us define the overlap decomposition.

**Definition 2.3.** A neural network $\Phi^l$ induces an equivalence relation that describes all regions of input space on which it is non-injective. It determines equivalence classes $[x] = \{y|\Phi^l(x) = \Phi^l(y)\}$ which together form the set,

$$\sim_{\Phi^l} = \{[x]|\Phi^l(x) = \Phi^l(y)\}. \tag{4}$$

The *overlap decomposition* describes the set of equivalence classes coming from the intersection source and is given by,

$$\mathcal{O}_{\Phi^l} = \{[x]|\Phi^l(y) \in \bigcap_{i \in I} \Phi^l(G_{J_i}^l) \text{ for } I \subset \{1, 2, ..., \# \text{ polyhedra}\}$$

$$\text{and } \Phi^l(y) \notin \bigcap_{i \in K} \Phi^l(G_{J_i}^l) \text{ for any } I \subsetneq K, \Phi^l(x) = \Phi^l(y), |I| > 1\}. \tag{5}$$

Note that the equivalence relation in equation 4 is reminiscent of a Reeb space (Edelsbrunner et al. (2008)), with the exception that we do not require that $x$ and $y$ lie on the same connected component $\Phi^{-1}(\Phi(x)) = \Phi^{-1}(\Phi(y))$. This definition is most useful for intuition and writing proofs. However, when finding overlaps algorithmically we use a coarser definition, as seen in equation 7, that produces identical results. This coarser decomposition can be fully computed given that we have knowledge of the polyhedral decomposition $\mathcal{G}^l$, by identifying the intersections between all pairs of different polyhedra and using a union-find structure on top (Kleinberg (2006)).

In the next section, we discuss how to compute this decomposition for a neural network and a supporting dataset. We also discuss how it can be leveraged to compute the homology groups of a neural representation. In order to build intuition, we show how a network can solve the XOR problem by using either a rank strategy or a overlap strategy in Appendix C.2.

## 3 Relating the Overlap Decomposition and Homology

### 3.1 Numerical Determination of the Overlap Decomposition through Linear Programming

We have stated that the overlap decomposition is a subset of the equivalence class $\sim_\Phi = \{[x]|\Phi(x) = \Phi(y)\}$. Here we show how to compute this set when working with finite data samples. As we prove later, under the assumption that the intersection between a polyhedron and the data manifold $P \cap \mathcal{M}$ is a convex set (this is automatically true if $\mathcal{M}$ is convex), we can use the polyhedral decomposition to exactly determine the overlaps between data samples. For two populated polyhedra, meaning there exist points in the dataset $\mathcal{D}$ that live in each of these polyhedra, the overlaps $\Phi_l(P_1) \cap \Phi_l(P_2)$ can be determined through linear programming. This is done by checking if given a point $p \in P_1$ there is another point $p' \in P_2$ such that $\Phi_l(p) = \Phi_l(p')$. This second point $p'$ can be found through linear programming under the constraints given by the H-representation of $P_1$ and $P_2$, which are given by $A_1 x \leq b_1$ and $A_2 x \leq b_2$ respectively. Thus, to check whether there is a point in $P_2$ that overlaps with $p \in P_1$, we get the following feasibility linear program,

$$\min_x (0^T x),$$

$$\text{Such that } A_2 x \leq b_2, \tag{6}$$

$$\Phi_{J_2}^l(x) = \Phi_{J_1}^l(p).$$

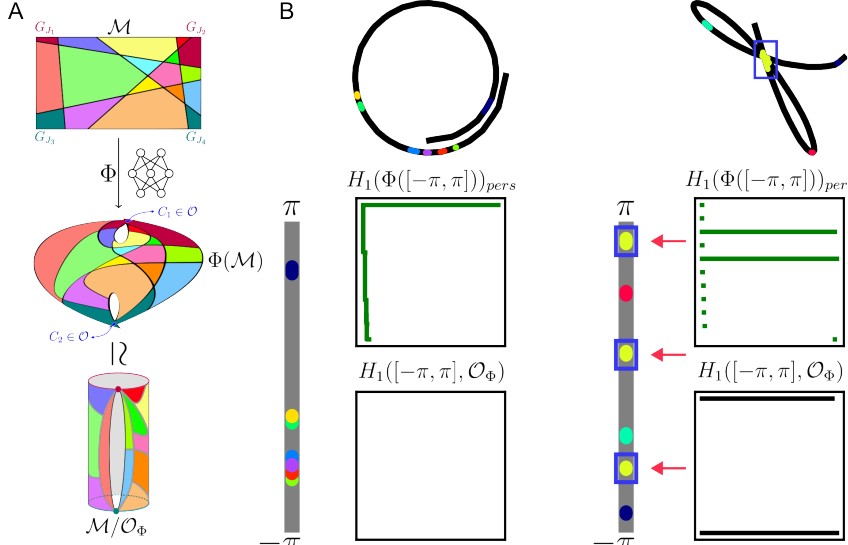

Figure 1: A) Illustration of the steps of our method. The manifold $\mathcal{M}$ is transformed by a network into a space on which the two corners are glued together. This space is homeomorphic to the punctured cylinder below it in which the corners $C_1$ and $C_2$ have been glued to a point. B) Visualization of our method, the top row shows two example non-linear curves, while the bottom shows their estimated homology groups through persistent homology (green) and quotient homology (black). Points that are glued through the overlap decomposition are given the same color and can be seen in the output space (top row) and the input space (the gray lines in the bottom row). As highlighted by the blue boxes in the second example, our method identifies the points that are glued by the neural network.

It is important to note that this approach looks for points within a polyhedron defined by the weights of a network, but this point does not need to be in the dataset $\mathcal{D}$. Therefore, this approach implicitly assumes that any point within $G_{J_2}^l \cap \mathbb{R}^{n_0}$ is a valid data point that *could have existed* in the dataset. The pseudocode for this is shown in Algorithm 1.

The advantage of this method is that it only identifies an overlap if there are two points on which the neural network generates the same output. In this sense, it is an exact determination of the overlap decomposition (up to errors due to machine precision) and avoids all issues that come with choosing an external metric. The drawback is that it requires the determination of the H-representation of many polyhedra and for high-dimensional input spaces it is prohibitively expensive to represent all of them. This could be improved through the use of other polyhedral representations that scale much better with dimension (Kochdumper and Althoff (2019); Sigl and Althoff (2023)). Another way to achieve a speed up is to precompute distances and skip looking for overlaps between points that are $\delta$ away from each other, in this work we always choose $\delta = 1$, except for section 4.2 where we set $\delta = 10$ for a slightly more conservative estimate. See Figure 10 in the Appendix to see how the algorithm scales with dimension and the effect of choosing different $\delta$'s.

### 3.2 QUOTIENT HOMOLOGY THROUGH THE OVERLAP DECOMPOSITION

Previously we stated that the rank and the overlap conditions form an exhaustive set of sources of topological change that a network can induce on the input manifold. This statement is formulated as a theorem B.2 and proven in the appendix. Here we state an intuitive version of our result.

**Theorem 3.1.** *(informal) The homology groups $H_k(\Phi(\mathcal{M}))$ are fully determined by the rank $\mathcal{R}_\Phi$ and the overlap $\mathcal{O}_\Phi$ decompositions.*

So far we have only described how to calculate the overlap decomposition. We have yet to describe a method to determine the rank decomposition. One suggestion is that we can treat regions in $\mathcal{R}_\Phi$ as contractible as in Beshkov and Einevoll (2024). As stated in that work, this is effective for regions of rank zero and one but can fail to produce an accurate result for regions of higher rank. Fortunately, it

turns out that if the intersections $\mathcal{M} \cap G_J^l$ are convex, then homology is invariant to the presence of low-rank maps and the overlap decomposition is sufficient to compute quotient homology, see B.5 for a proof. This leads to the second fundamental theorem of this work.

**Theorem 3.2.** *Given a neural network $\Phi$ with a polyhedral decomposition $\mathcal{G}_J^l$ s.t. $\mathcal{M} \cap G_J^l$ is convex for any $G_J^l \in \mathcal{G}_J^l$, there is an isomorphism $H_k(\Phi(\mathcal{M})) \simeq H_k(\mathcal{M}/\mathcal{O}_\Phi)$.*

An illustration of this theorem can be seen in panel A of Figure 1. While the assumption that the data lives on a convex set is very restrictive, it is important to emphasize that our assumption is much weaker. We only require that the intersections between the data manifold and the polyhedra induced by the structure of a network are convex. Given a large enough network in which the number of polyhedra grows polynomially in the number of neurons (on the order of $(T\#\text{neurons})^{n_0}/n_0!$, with $T$ a positive constant, as stated in Hanin and Rolnick (2019)) this condition will obtain, except for fractal data. This assumption also turned out to be empirically true in the simulations performed in this paper and the additional convexity analysis found in Figure 9 of the Appendix.

## 4 RESULTS

### 4.1 COMPARISON BETWEEN QUOTIENT AND PERSISTENT HOMOLOGY

Persistent homology is an essential tool for computing homology groups of data from different domains of science (Carlsson (2009); Wasserman (2018)), including machine learning (Papamarkou et al. (2024)). It is not a purely topological method as it also tracks geometric features of the underlying point cloud (Bubenik et al. (2020); Turkes et al. (2022)). Plus, it is highly sensitive to the sampling density and the chosen metric. While sometimes there are ways to overcome these problems, it is likely that geometric properties are distorting the study of representational topology.

To show the effectiveness of our method, we generated datasets of highly non-linear curves by the equation $f(\theta) = [\cos(a\theta)\cos(b\theta), \cos(a\theta)\sin(b\theta)]^T$. The parameters $a, b$ were randomly sampled from a uniform distribution on the interval $[-1, 1]$. We sampled 500 equally spaced points $\theta \in [-\pi, \pi]$. Afterwards, we used an MSE loss to optimize a neural network with three layers, each with a width of 50 neurons, to predict these functions given the input points $(\theta, 0)$. The parameters for all simulations are described in detail in Appendix D. Following training we compute the first homology groups $H_1$ of the output of the final network layer. We do this in two ways: standard persistent homology with the Euclidean metric and quotient homology using the metric as described in Appendix A.5.1.

In Figure 1 we show two examples of learned curves and their associated homology groups given by the barcodes below. One can see that in the first example, standard persistent homology identifies a circular feature as the ends of the interval are close in the output despite the fact that the network does not map them onto each other. This error is avoided when using quotient homology, as seen by the lack of bars in the barcode plot. In the second example, we see that both persistent homology and quotient homology generate two persistent features. However, given that we have access to the overlap decomposition, we are also able to track exactly which points were glued together by the network and were therefore responsible for the generation of the two circles. Many other examples are shown in Appendix 8.

### 4.2 REVISITING MANIFOLD PROPAGATION THROUGH NEURAL NETWORKS

Previous work by Naitzat et al. (2020) has studied how the homology groups of different toy datasets evolve through the layers of an almost perfectly trained neural network. In order to estimate the distances between points, they generate a $k$-nearest neighbor graph, with $k$ being optimized over the input data, and apply a shortest path algorithm on top of this graph. Following this, they calculate the homology groups at a particular scale for which they are stable and take that as an estimate of the homology groups of the neural representation at a layer.

They observe that the initial layers of a network rapidly reduce the Betti number of the input manifold in all three datasets. Given that persistent homology also provides geometric, rather than purely topological information (Bubenik et al. (2020)), we check whether our approach, which is only concerned with topological information, agrees with their results. Since working with the amount of samples used in the datasets generated by Naitzat et al. proved to be too computationally heavy for

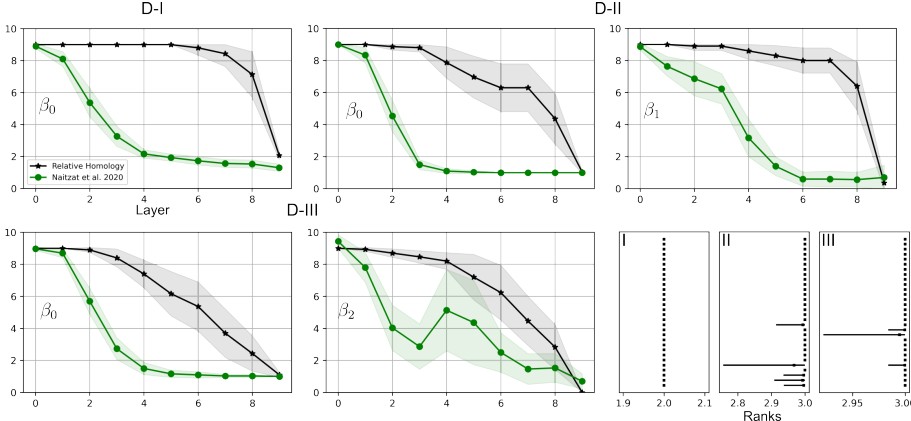

Figure 2: Reproduced Betti numbers from Naitzat et al. (green) and a quotient homology calculation (black), with shaded error-bars clipped to 9, which was the maximum observed value. In all three datasets we see a slower decay of all topological features when quotient homology is used. The three plots in the bottom right show distributions of the ranks generated by a neural network, indicating that the rank decomposition is unlikely to play a role in the estimated curves.

the computation of the polyhedral and overlap decompositions, we sampled 7800, 7500, and 8000 points for the three datasets, respectively and reran all parts of their analysis (see Appendix D for the full details of our implementation). Despite using smaller datasets, we managed to reproduce the quickly decaying Betti numbers observed in their work.

As can be seen in Figure 2, quotient homology shows different behavior than the reproduced curves (one might also compare them to the original curves in their paper for a similar conclusion). Thus, our analysis shows that if we consider purely topological transformations, then it seems that while neural networks might initially twist the manifolds to look like they have lost topological features, actual changes in the topology of such manifolds happen much more gradually. This leads one to wonder whether topological changes in the sense of non-homeomorphism or rather strong geometric changes matter more for network performance. We focus more on this question in the discussion.

The datasets in Naitzat et al. are not convex, and if low-rank regions exist we cannot apply Theorem 3.2 without checking for the convexity of each intersection, as then we would miss changes in topology induced by the rank source. To check if this was in fact an issue, we computed the rank of the maps over each polyhedron in the polyhedral decomposition for all models and datasets. The results can be seen in the bottom right plot of Figure 2 and show that regions of lower rank appear very rarely and are unlikely to have a severe impact on the curves that we estimate.

### 4.3 OVERLAP DECOMPOSITIONS IN RANDOM AND TRAINED NETWORKS

So far we have shown that by using a quotient homology approach we can study the topology of neural representation in neural networks by knowing the input space and the overlap decomposition. This has helped us separate the topological from the geometric features of a neural representation. Assuming that a network manages to learn a function almost perfectly, we would expect that the topology it generates is going to be similar to the topology that is induced by the function being learned. Classification problems inherently have the property of simplifying the topology of the initial data (a large part of the data is sent to a single point representative of a class) (Papyan et al. (2020); Rangamani et al. (2023)).

However, even before any learning occurs, we would like to have a good initialization of the network. Thus, it is important to understand the impact of training on the overlap decomposition. The size of the overlap decomposition at initialization should reflect the ability of a network to glue different pieces of the input. Therefore, the number of overlap regions is a measure of expressivity reflecting the degree to which a network can implement non-injective functions.

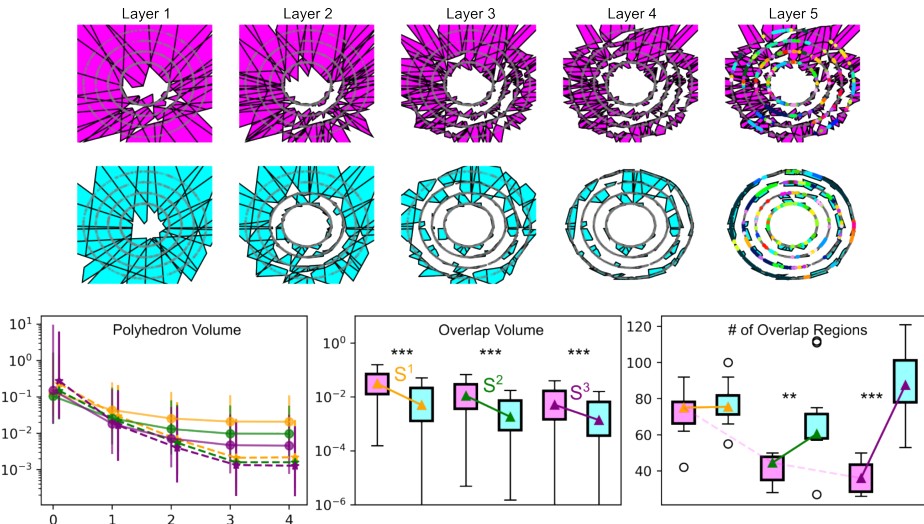

Figure 3: **(Top row)** Visualization of the polyhedral decomposition at initialization (magenta) and after training (cyan) across layers. Points of the same color belong to the same overlap class and the gray points do not overlap. In the last layer we see the points in the overlap decomposition and note that some of them overlap with each other both before and after training. **(Bottom row)** (left) The dimension of the spheres is color coded with $S^1$-orange, $S^2$-green and $S^3$-purple. Polyhedron volume decreases across layers and dimensions before (circles) and after (stars) training. (center) Overlap volume decreases after training. Stars indicate Bonferroni corrected significance using a Kruskal-Wallis test (Kruskal and Wallis (1952)). (right) The number of overlap regions increases after training.

To study this further, we create another toy problem in which we can compare overlap regions before and after training. We take four d-spheres with radii $\{1, 1.5, 2, 2.5\}$ respectively and assign a class of 0 to the spheres with radius $\{1, 2\}$ and a class of 1 to the other two. We do this for spheres of dimension $\{1, 2, 3\}$ by sampling points i.i.d. from a normal distribution in $d + 1$ and afterwards projecting them to the sphere of the desired radius through normalization. We sample 500 points from each such sphere and train 10 Kaiming-initialized (He et al. (2015)) networks for each dimension. Each network has an architecture of $\{d+1, 25, 25, 25, 25, 2\}$ and is optimized for 1000 epochs with a learning rate of 2e-5.

Before and after training, we computed the polyhedral and overlap decompositions through linear programming. A plot of the regions generated by one network at each layer, before (magenta) and after (cyan) training, is shown in Figure 3. As one can see, points fall in the overlap decomposition only in the last layer of the network. Another impression that this plot makes is that the polyhedra in the trained network seem to have a smaller volume. The plots confirm that this is the case both for those that participate in the overlap decomposition as well as those that do not.

Finally, since classification problems are likely more non-injective than an arbitrary function, one might expect that the number of overlap regions increases as a result of training. Looking at the final plot in Figure 3, this does seem to be the case, although this trend is only significant for the datasets using two- and three-dimensional spheres. We achieve similar results when using an orthogonal initialization; see Figure 11 in the Appendix.

## 5 DISCUSSION

While deep neural networks have proved to be extremely powerful and flexible, our theoretical understanding of their inner workings is still lacking. In this work we have related them to piecewise-linear functions over a set of convex polyhedra and described the areas of the input domain on which they act non-injectively. We further applied this description to define a quotient homology theory,

which we then used to compute homology groups that are sensitive purely to topological rather than geometric features. We now discuss some situations in which our approach will fail.

## 5.1 HOMOLOGICAL TYPE 1 AND TYPE 2 ERRORS

While in a perfect world we would like to identify all polyhedra that a network generates, due to combinatorial explosion, it is not feasible to do this for large networks. One way to do this would be to list all possible codewords that could specify a polyhedron in $\mathcal{G}^l$ and then find their H-representation. This means that we need to identify $2^{\#\text{neurons}}$ polyhedra for each layer. For this reason, we have focused on only considering polyhedra that are populated by at least one point in a dataset. This means that for a populated polyhedron $p \in P_i$ where the network implements a map $\Phi|_{P_i}$, there might be another unpopulated polyhedron $P'$ such that $\Phi|_{P_i}(p) = \Phi|_{P'}(z)$, where $z \in P'$. This would be a gluing that the network performs on the input space that is missed due to the limited dataset and is reminiscent of a homological type 2 error.

Another issue arises from the fact that if we think of neural networks as functions between vector spaces $f : \mathbb{R}^{n_0} \to \mathbb{R}^{n_L}$, then the polyhedral decomposition they define extends to all of input space regardless of the actual manifold that a dataset comes from. Since we tend to embed input data in large vector spaces, there might be a point $z \notin \mathcal{M}$ within one of the populated polyhedra, such that $\Phi|_{P_i}(p) = \Phi|_{P_j}(z)$. This would imply that the network applies a gluing when it does not do that on the manifold $\mathcal{M}$ from which the data is sampled. This is reminiscent of a homological type 1 error.

## 5.2 COMPUTATIONAL COMPLEXITY OF OVERLAP DETECTION

We have proposed an algorithm for the discovery of the overlap decomposition. While an exact quantification of its computational complexity is not within reach, here we discuss the relation between its tractability and network or data structure.

The first part of our algorithm relies on identifying the H-representation of each polyhedron. It is known that in a worst-case scenario the number of polyhedra in a neural network grows exponentially in depth and polynomially in width (Goujon et al. (2024)). However, the expected number of polyhedra seems to grow polynomially in the number of neurons (Hanin and Rolnick (2019)) with the order depending on the input dimension. Furthermore, since we only consider polyhedra that are populated by data, their number is further bounded by the number of samples in a dataset. Together, these facts imply that the computation and storage of H-representations might be less of a problem than worst case analysis might imply. This is also supported by the simulations in Figure 10. In the case that the number of polyhedra is still restrictive, further work leveraging the computational efficiency of alternative representations such as the Z and M representation (Kochdumper and Althoff (2019); Sigl and Althoff (2023)) is needed.

The second part of our algorithm requires determining whether overlaps exist through linear programming. In principle, if $n$ are the average number of points in a polyhedron and $N$ is the number of polyhedra, one would need to solve on the order of $N(N-1)n^2$ linear programming tasks. Since points that are far away in representation space are unlikely to be glued, this can be significantly improved by only considering overlaps when their distance is below some threshold $\delta$, see Figure 10. The computational complexity of interior point methods, such as the HiGHS solver used in our analysis, in linear programming is polynomial in the number of variables and constraints, as shown by Karmarkar's algorithm (Karmarkar (1984)). Since both the number of overlaps and the complexity of linear programming scale polynomially, this second step also scales polynomially.

Despite the fact that both steps have expected polynomial scaling, given the sheer size of modern neural networks, computing the overlap decomposition can still be prohibitively expensive. Here we have presented two approaches for improving this related to polyhedral representations and distance thresholding. Given the novelty of our approach, finding more efficient methods that allow us to probe more advanced neural architectures is an exciting avenue of future research.

## 5.3 TOPOLOGICAL VERSUS GEOMETRIC TRANSFORMATIONS

We showed that previous calculations of Betti numbers in neural networks have likely described geometric rather than topological features and that topological changes occur much more gradually

in neural representations across the layers of a network. This naturally raises the question of which type of transformation we should focus our efforts on understanding further. As argued in Petri and Leitão (2020), from the perspective of a linear classifier, the topology of the two classes is irrelevant as long as they are linearly separable in the final layer. This supports the view that when it comes to topology, only that of the decision boundary carries significance.

It is important to note that, while the geometric view makes intuitive sense, it fails to capture the behavior of actual networks. A well-known phenomenon in deep networks trained with cross-entropy on classification tasks is neural collapse (Papyan et al. (2020); Rangamani et al. (2023)), in which samples collapse to their class mean. This is an example of topological simplification that occurs despite it not being necessary from the perspective of a classifier. Furthermore, not all problems that we are interested in are classification problems. Certain tasks will require the implementation of topological changes (a trivial example is learning a constant function, which glues everything to a point). In such cases, it is important to be able to describe them without interference from geometric sources. We are hopeful that the tools introduced in this paper help us get closer to this possibility.

### 5.4 NUMBER OF OVERLAPS AS A MEASUREMENT OF EXPRESSIVITY

Expressivity describes the complexity of the functions that a network architecture can implement and is often measured by counting linear regions. Here we have argued that a related measure is the number of regions in the overlap decomposition. This notion of expressivity describes the degree to which a network can implement non-injective maps. We have not thoroughly evaluated overlap expressivity in this work, but counting the number of overlap regions at initialization shows that compared to the number of polyhedra, they might scale in a non-strictly monotonic fashion as width and depth increase; see Figure 12 in the Appendix.

In principle, it is possible that certain network architectures might have many linear regions but fail to be able to implement highly non-injective maps. As we show in B.6, this is the case for networks with a single layer. We also establish a lower bound on the number of overlaps for networks with $(1, N, 1)$ architecture in C.3. The features that determine whether a network has the ability to generate such regions in more complex architectures are still unknown. Furthermore, since as we show in Theorem B.2 and with an example C.2, non-injectivity can be achieved through both the rank and the overlap decomposition. Understanding how having more linear regions in deeper networks corresponds to the capacity of implementing more non-injective functions through the rank or overlap decomposition is an open question that will hopefully lead to a better understanding of neural networks and their inner workings.

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

# A    PREREQUISITES

Here we provide some prerequisite knowledge which will hopefully come in handy to readers unfamiliar with some of the concepts that we have used.

## A.1    POLYHEDRAL GEOMETRY

The atomic elements of this work are convex polyhedra. There are two popular ways to represent such objects. The first is called the H-representation and is given by the intersections of half-spaces. The second is called the V-representation and is given by the convex hull of a set of points. Here we will work purely with the H-representation, although previous work has also used V-representations (Masden (2022)).

**Definition A.1.** A half-space is the set given by a linear equation,

$$H_- = \{x | a^T x \le b\}.$$

In the H-representation, a polyhedron is described as the intersection of finitely many half-spaces. Then, a system of linear equations specified by a matrix $A$ and a vector $b$ form the H-representation of a polyhedron,

$$P = \{x | Ax \le b\}.$$

It is worth noting that if we replace the inequality in the half-space definition with a strict equality, we get hyperplanes defined by the equations $a_i^T x = b$, where $a_i$ is the i-th row of the matrix $A$. Such hyperplanes are called the **supporting hyperplanes** of the polyhedron. Several hyperplanes also define a **hyperplane arrangement** which is in **general position** whenever $\dim(H_1 \cap ... \cap H_n) = m - n$, where m is the dimension of the input space and $m \ge n$. In the case where $m < n$, the intersection should be empty for an arrangement to be in general position. In this work we will only concern ourselves with arrangements in general position, which are known to almost always occur in neural networks (Grigsby and Lindsey (2022)).

One property of such polyhedra that we will use later is the fact that they are always convex. This will come in handy when we prove the conditions under which only the overlap decomposition is necessary when computing quotient homology. For all calculations using polyhedra we used the **polytope** package in python (https://github.com/tulip-control/polytope) and specified a bounding box of $[-100, 100]$ around every input space.

## A.2    LINEAR PROGRAMMING

While the field of neural networks is still in its infancy, there are well established ways to study hyperplane arrangements. One such approach which we use to discover overlaps is linear programming. A linear program is a linear optimization technique that finds a solution $x$ under a set of linear constraints,

$$\max_x (c^T x),$$
$$\text{Such that } Ax \le b.$$

A special type of linear program which we mainly concern ourselves with in this work is the discovery of a feasible region. In this case we just want to find any point $x$ that satisfies the inequality constraints. This can be computed by substituting $c$ with a zero vector and thereby only looking for a point $x$ that satisfies the linear constraints. Linear programs can be solved through a variety of methods, in this work we have used the **scipy** wrapper of the HiGHS solver (Huangfu and Hall (2018)).

## A.3    SIMPLICIAL COMPLEXES AND FILTRATIONS

Data typically comes in the form of a point cloud in which there is no apriori knowledge of the underlying structure of the space from which it is sampled. In order to study the topology of neural representations across the layers of a network, one has to associate some structure to these points.

Given a metric space $X$, a standard approach for this is the Vietoris-Rips complex, defined as the set $V_\epsilon(X) = \{S : d(x_i, x_j) \leq \epsilon, \forall x_i, x_j \in S\}$. This forms an abstract simplicial complex in which every $S$ with $k$ elements is a $(k-1)$-simplex. Given this strucutre we can compute simplicial homology.

Very often we do not have knowledge of the right value of $\epsilon$ and therefore use the strategy of persistent homology. In this case one builds a filtration of simplicial complexes, known as the Vietoris-Rips filtration. This is done by progressively increasing the value of $\epsilon$ . Since we know that $V_\epsilon \subseteq V_{2\epsilon}$, this corresponds to adding new simplices to each previous simplicial complex.

## A.4 QUOTIENT MAPS

The main intuition behind our approach is that a network can change the topology of an input manifold by gluing different pieces of it together. This intuitive idea is studied in topology through the concept of a quotient space under an equivalence relation $\sim$. Quotient spaces are central in all following proofs and therefore deserve more attention. The first concept that needs clarification is that of an equivalence relation.

**Definition A.2.** A relation $\sim$ is an equivalence relation if and only if obeys the following properties.

- $x \sim x$ (reflexivity).

- $x \sim y \Leftrightarrow y \sim x$ (symmetry).

- If $x \sim y$ and $y \sim z$ then $x \sim z$ (reflexivity).

If we are given a map $f : X \to Y$, it is often the case that $f$ produces the same output on several values of $X$ and therefore splits $X$ into several equivalence classes denoted by $[x] = \{z \in X | f(z) = f(x)\}$ (note that points on which $f$ is unique generate an equivalence class with a single point $[x] = \{x\}$). Together these equivalence classes can be written as a set $X/\sim_f = \{[x] | x \in X\}$ and there is *canonical quotient map* $q : X \to X/\sim_f$ sending any $x$ to its equivalence class $[x]$. This set is then equipped with the *quotient topology*, meaning that any subset $U \subseteq X/\sim_f$ is open if and only if $q^{-1}(U)$ is open in $X$.

## A.5 RELATIVE HOMOLOGY

While standard homology calculations are at this point quite familiar to machine learning researchers, the concept of relative homology is still rather unexplored in the machine learning literature. Intuitively, one of the main motivations behind relative homology is to understand how the homology groups of a space change as we glue a subspace from it. This is possible as long as $X$ and $S$ are compact Hausdorff and there is some neighborhood of $S$ which is a strong deformation retract to some closed neighborhood of $S$ (see **Theorem 2.14** and **Corollary 2.15** in Vick (2012)). Therefore, given this mild condition, we can use relative homology to study the homology groups of quotient spaces $X/S$.

The precise steps to calculate relative homology work as follows. If we have a simplicial complex of a space $V_\epsilon(X)$ we can associate a chain complex $C_k(X)$ containing all k-chains. Furthermore, we can associate a chain complex $C_k(S)$ to a subcomplex $S \subset X$. Finally, we also need to define a chain complex structure on the quotient $C_k(X, S) = C_k(X)/C_k(S)$. These objects fall inside of a short exact sequence,

$$0 \longrightarrow C_k(S) \xrightarrow{i} C_k(X) \xrightarrow{j} C_k(X, S) \longrightarrow 0,$$

where $i$ is the inclusion $S \hookrightarrow X$, and $j$ is the quotient map $j : X \to X/S$.

What we would really like to have is a sequence between the quotients of the chain groups through some sort of boundary map like those used in standard homology calculations,

$$... \longrightarrow C_k(X, S) \xrightarrow{\partial_k} C_{k-1}(X, S) \xrightarrow{\partial_{k-1}} C_{k-2}(X, S) \longrightarrow ...$$

Given such a map, we can define homology groups like usual,

$$H_k(X, S) = \ker \partial_{k-1}/\mathrm{Im}\partial_k.$$

While we cannot get such a map directly, we can use the zig-zag lemma (Munkres (2018)) on the short exact sequence of chain groups to derive a long exact sequence of homology groups, which we can use to compute relative homology groups given knowledge of $H_k(S)$ and $H_k(X)$.

$$... \longrightarrow H_k(S) \xrightarrow{i_*} H_k(X) \xrightarrow{j_*} H_k(X, S) \xrightarrow{\partial_*} H_{k-1}(S) \longrightarrow ...$$

The star subscript denotes that the maps are induced by the homology functor $H_k : f \to f_*$. This new boundary map connecting the homology groups $H_k(X, S)$ to $H_{k-1}(S)$ is called the *connecting homeomorphism* (for details see Hatcher (2002) pages 115-119).

Under the mild conditions stated above, relative homology is equivalent to the homology of the quotient space $X/S$ when $S$ is a subspace of $X$. However the overlap decomposition that we describe actually contains many and not just one such subspaces. Since all of these subspaces are disjoint, we can sequentially stack relative homology sequences to find out the final homology of the quotient space $\mathcal{M}/\hat{\mathcal{O}}$ (note that we use the coarser definition of the overlap decomposition 7 to avoid taking an infinite amount of steps in the sequence). This leads to the sequence,

$$... \xrightarrow{\partial^*} H_k(S_1) \xrightarrow{\iota^*} H_k(\mathcal{M}) \xrightarrow{j^*} H_k(\mathcal{M}, S_1) \xrightarrow{\partial^*} ...$$

$$... \xrightarrow{\partial^*} H_i(S_2) \xrightarrow{\iota^*} H_k(\mathcal{M}, S_1) \xleftarrow{j^* \, id} H_k(\mathcal{M}/S_1, S_2) \xrightarrow{\partial^*} ...$$

$$... \xrightarrow{\partial^*} H_i(S_3) \xrightarrow{\iota^*} H_k(\mathcal{M}/S_1, S_2) \xleftarrow{j^* \, id} H_k((\mathcal{M}/S_1)/S_2, S_3) \xrightarrow{\partial^*} ...$$

where $S_i \subset \mathcal{M}$ are ordered elements of $\hat{\mathcal{O}}$. There is a finite number of them since a network with finite neurons can generate a finite number of polyhedra and therefore there can only be a finite number of intersections between them. While we do not explicitly calculate relative homology groups in our experiments, this relation between relative and quotient homology can be useful in future work as it provides a way to apply exact sequences to the study of the topology of neural representations.

### A.5.1 COMPUTING QUOTIENT AND RELATIVE HOMOLOGY IN DATA

In the settings considered in this work, computing quotient homology can be thought of as sequentially computing relative homology groups. The fact the computation of relative homology requires knowledge of the homology groups $H(X)$ and $H(S)$, brings both advantages and disadvantages. On the bright side, it is sometimes the case that we might know something about the structure of the input data, whereas studying the output of a highly non-linear function like those implemented by neural networks is much more difficult. In this way, quotient homology avoids having to deal with the non-linear effects generated by neural networks. The downside of this approach is that if we do not know the homology groups of the input data (which is often the case), we need a way to estimate them. So how should we approach this?

If we have a reasonable guess for the spatial scale at which the input data can be analyzed, we can fix some $\epsilon$ and use the Vietoris Rips complex at that value to calculate homology groups at a single point. Then to calculate quotient homology we can append a single point to each simplex in $S$. This makes each simplex $\sigma \in S$ into a contractible cone and therefore homeomorphic to taking the quotient $X/S$.

Alternatively, we can still use persistent homology but set the distances between every two points $s_i, s_j \in \sigma$ to 0. After this we can define the quotient (pseudo)metric with $p_1 \in [x]$ and $q_n \in [y]$ by,

$$d([x], [y]) = \inf\{d(p_1, q_1) + d(p_2, q_2) + ... + d(p_n, q_n)\}.$$

This effectively computes the shortest path between two points given that the distances within each equivalence class $[x]$ and $[y]$ are set to 0. Computationally, we implement this by first setting the

aforementioned distances to 0 and afterwards, computing the shortest path on the distance weighted adjacency matrix using Djikstras algorithm (Dijkstra (1959)). While we developed an implementation of the first approach, it proved much more computationally efficient to use the second.

## B   PROOFS

We start this section by showing that the only source of topological change that a network can implement is due to a failure to be injective. While such maps can fail to be surjective if we define $\Phi : \mathcal{M} \to \mathbb{R}^n$, since the codomain can fail to be covered, we are more interested in how the data manifold is transformed. Therefore, for the purposes of this goal we will define the codomain of the map to be $Im(\Phi)$, where $Im(\Phi) := \{\Phi(x)|x \in \mathcal{M}\} \subseteq \mathbb{R}^n$. This makes the mapping trivially surjective. Furthermore, we omit the layer index as our proofs apply to any stage of the network.

**Theorem B.1.** *A ReLU neural network $\Phi : \mathcal{M} \to Im(\Phi)$, with $\mathcal{M}$ being a compact manifold, is not a homeomorphism iff $\Phi$ is not injective.*

*Proof.* It is useful to remind the reader of the properties necessary for a map to be a homeomorphism.

- $\Phi$ is bijective,

- $\Phi$ is continuous,

- There is an inverse continuous function $\Phi^{-1}$.

From this we can see that since if $\Phi$ is not injective it is therefore not bijective, this is enough to prove one direction of the statement. Now we have to show that if $\Phi$ is not a homeomorphism then it is not injective. We will do that by showing that all other properties for a homeomorphism are guaranteed by the nature of the maps that neural networks can implement. For this we use **Theorem 2.1** in Arora et al. 2018., which states that every ReLU deep neural network represents a continuous piecewise linear function.

Since such neural networks are continuous piecewise linear functions, the continuity of $\Phi$ follows. We have seen that surjectivity trivially follows from the way we define the codomain, so the only thing left to show is that $\Phi^{-1}$ is also continuous.

It turns out that such a function need not be continuous, but that occurs solely when $\Phi$ is not injective. Let us assume that $\Phi$ is injective, then it is a continuous bijection (surjectivity applies everywhere). Since $\mathcal{M}$ is compact we can use **Theorem 26.6** from Munkres (2000) to see that this implies that $\Phi^{-1}$ is continuous (the theorem in Munkres also requires that $Im(\Phi)$ is Hausdorff but this property follows from the fact that $\mathcal{M}$ is a compact manifold and $\Phi$ is continuous). This proves that injectivity implies that we have a homeomorphism and thus all topological changes occur as a result of the network failing to be injective. $\square$

Next we show that the non-injectivity of neural networks can be described by two sources related to the local rank of a map and the overlap of regions in the polyhedral decomposition.

**Theorem B.2.** *Denote the set of equivalence classes generated by $\Phi$ as $\sim_\Phi = \{[x]|\Phi(x) = \Phi(y)\}$. This set is contained in the union of $\mathcal{R}_\Phi = \{[x]|rank(\Phi|_x) < \dim(\mathcal{M}), y \in G_J, \Phi(x) = \Phi(y)\}$ and $\mathcal{O}_\Phi$ (defined in 2.3).*

*Proof.* Any equivalence class $[x]$ in $\sim_\Phi$ contains a set of points in $\mathcal{M}$. The input manifold has a polyhedral decomposition and can be written as $\mathcal{M} = \bigcup_J G_J$, what this means is that the set of points contained in $[x]$ live in either one or several regions $G_J$. Thus, depending on how many polyhedra are occupied by the points in $[x]$, we can consider two cases.

- **Case 1:** The points in $[x]$ all fall in the same region $G_J$. We know that within the same region the network applies an affine transformation and whenever such a transformation is full rank, the map is locally a homeomorphism. Therefore, for a map to be non-injective within a single region, it has to fail to be full rank. This is the exact set described by the rank decomposition $\mathcal{R}_\Phi$.

- **Case 2:** The points in $[x]$ fall in several different regions $G_J$. In this case we know that since for any two points in $[x]$, coming from different regions $G_J$, we have $\Phi(x) = \Phi(y)$

then $\Phi(y) \in \bigcap_{i \in I} \Phi(G_{J_i})$ for any $y \in [x]$. This set exactly coincides with the overlap decomposition $\mathcal{O}_\Phi$.

Since these are the only two possibilities for where the points in any $[x]$ can come from, we have shown that $\sim_\Phi \subset \mathcal{R}_\Phi \cup \mathcal{O}_\Phi$. The reason we do not have an equality is that it is possible that there are two equivalence classes $[x]$ and $[y]$ in $\mathcal{R}_\Phi$ that fall in a larger equivalence class $[x]_\mathcal{O} \in \mathcal{O}_\Phi$. In this case, $\sim_\Phi$ will contain $[x]_\mathcal{O}$, but not the other two classes. We can also describe such an equality if we define a pairwise intersection operation as,

$$\mathcal{R}_\Phi \cap_{pair} \mathcal{O}_\Phi = \{A \cap B | A \in \mathcal{R}_\Phi, B \in \mathcal{O}_\Phi\}.$$

This expression is strictly a subset $\mathcal{R}_\Phi \cap_{pair} \mathcal{O}_\Phi \subset \mathcal{R}_\Phi$ and it filters out all equivalence classes in $\mathcal{R}_\Phi$ that are subsets of the classes in $\mathcal{O}_\Phi$. Then we can state $\sim_\Phi = \mathcal{R}_\Phi \cup \mathcal{O}_\Phi - \mathcal{R}_\Phi \cap_{pair} \mathcal{O}_\Phi$. $\quad\square$

Next we show that if we treat $\Phi$ as a canonical quotient map and we have knowledge of the rank and overlap decompositions, then we can compute homology groups through quotient homology. This is a corrollary of the fact that the image of $\Phi$ is homeomorphic to the canonical quotient map induced by the equivalence relation $\sim_\Phi$.

**Theorem B.3.** $\Phi(\mathcal{M})$ *is homeomorphic to* $q(\mathcal{M})$. *Where* $q : \mathcal{M} \to \mathcal{M}/\sim_\Phi$ *is the canonical quotient map.*

*Proof.* We essentially have the following diagram:

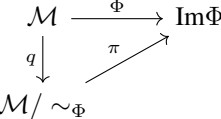

If we can find some $\pi$ that is a homeomorphism, then we can write $\Phi = \pi \circ q$ and the proof would be complete. We can construct such a map in the following way,

$$\pi([x]) = \begin{cases} \Phi(x) \text{ if } |[x]| = 1, \\ \Phi(y) \text{ for some } y \in [x]. \end{cases}$$

This function is injective, since $\pi([x]) = \pi([z])$ implies $\Phi(x) = \Phi(z)$ and therefore $[x] = [z]$. It is also surjective as for any $z \in \text{Im}\Phi$, there is an $x \in \mathcal{M}$ such that $\Phi(x) = z$ and $\pi([x]) = z$. The continuity of $\pi$ simply follows from the fact that $\Phi$ is continuous. Finally, the continuity of the inverse $\pi^{-1}$ again follows from **Theorem 26.6** in Munkres (2000) as the quotient space of a compact space is compact, and the precise choice of $y$ is irrelevant. $\quad\square$

**Corollary B.4.** *The homology groups* $H_k(\Phi(\mathcal{M}))$ *are equivalent to the quotient homology groups* $H_k(\mathcal{M}/\sim_\Phi)$.

*Proof.* This follows simply from the fact that homeomorphic spaces have identical homology groups. $\quad\square$

It turns out that in cases for which $\mathcal{M} \cap G_J$ is convex for any $J$, we can ignore the rank decomposition and find the quotient homology groups only given knowledge of the overlap decomposition which we have a way to compute. This is proven in the following theorem.

**Theorem B.5.** *If* $\mathcal{M} \cap G_J$ *is convex* $\forall G_J$, *then* $H_k(\Phi(\mathcal{M})) \simeq H_k(\mathcal{M}/\mathcal{O}_\Phi)$.

*Proof.* In the corrolary of the previous theorem we have already shown that $H_k(\Phi(\mathcal{M})) \simeq H_k(\mathcal{M}/\sim_\Phi)$. We also know that $\sim_\Phi = \mathcal{R}_\Phi \cup \mathcal{O}_\Phi - \mathcal{R}_\Phi \cap_{pair} \mathcal{O}_\Phi$, therefore our proof will be complete if we can show that quotiening out the part of the rank decomposition that does not

intersect the overlap decomposition leaves the homology groups invariant. We will abuse notation by writing $X - A$ instead of $X - \bigcup A$ to denote the points in $X$ that do not fall in any of the sets in $A$.

Since we know that homology groups are invariant under homotopy equivalence, we require that $\mathcal{M}/\sim_\Phi$ is homotopy equivalent to $\mathcal{M}/\mathcal{O}_\Phi$. This is represented by the diagram:

$$
\begin{array}{ccc}
\mathcal{M} & \xrightarrow{\ \ p\ \ } & \mathcal{M}/\mathcal{O}_\Phi \\
{\scriptstyle q}\downarrow & {\scriptstyle\pi}\ \ {\scriptstyle\sigma} & \\
\mathcal{M}/\sim_\Phi & &
\end{array}
$$

The two arrows $p$ and $q$ coming out from $\mathcal{M}$ are canonical quotient maps. To show homotopy equivalence we now prove that $\pi \circ \sigma$ and $\sigma \circ \pi$ are homotopic to $\mathrm{id}_{\mathcal{M}/\mathcal{O}_\Phi}$ and $\mathrm{id}_{\mathcal{M}/\sim_\Phi}$ respectively. The two quotient maps split $\mathcal{M}$ into equivalence classes that we denote as $[x]_p$ and $[x]_q$. Since both $p$ and $q$ take a quotient with respect to $\mathcal{O}_\Phi$ the equivalence classes $[x]_q$ and $[x]_p$ will match on $x \in \mathcal{M} - (\mathcal{R}_\Phi - \mathcal{R}_\Phi \cap_{pair} \mathcal{O}_\Phi)$. Therefore on this set $\pi$ and $\sigma$ map identical equivalence classes to each other. The interesting part of the proof is dealing with the remaining set that is generated purely by the rank decomposition.

We can define the map $\sigma : \mathcal{M}/\mathcal{O}_\Phi \to \mathcal{M}/\sim_\Phi$ as another quotient canonical map such that $q = \sigma \circ p$, that collapses the points that were missed by $p$ or,

$$
\sigma([x]_p) = \begin{cases} q(x) \text{ if } x \in \mathcal{M} - (\mathcal{R}_\Phi - \mathcal{R}_\Phi \cap_{pair} \mathcal{O}_\Phi), \\ q \circ p^{-1}([x]_p) \text{ if } x \in \mathcal{R}_\Phi - \mathcal{R}_\Phi \cap_{pair} \mathcal{O}_\Phi. \end{cases}
$$

Note that since $p$ is injective on points that are not in $\mathcal{O}_\Phi$, it is invertible on this subdomain. The map $\pi : \mathcal{M}/\sim_\Phi \to \mathcal{M}/\mathcal{O}_\Phi$ is tricker, but we define it piecewise as,

$$
\pi([x]_q) = \begin{cases} p(x) \text{ if } x \in \mathcal{M} - (\mathcal{R}_\Phi - \mathcal{R}_\Phi \cap_{pair} \mathcal{O}_\Phi), \\ [z]_p \text{ such that } \sigma([z]_p) = [x]_q \text{ for some } z \in \mathcal{R}_\Phi - \mathcal{R}_\Phi \cap_{pair} \mathcal{O}_\Phi. \end{cases}
$$

For the second condition we need to choose some representative $[z]_p$, since there are many $\sigma([z]_p) = [x]_q$, this particular choice does not really matter as in the end $\sigma \circ \pi([x]_q) = \sigma([z]_p) = [x]_q$ and $\pi \circ \sigma([x]_p) = \pi([x]_q) = [z]_p$. The first composition of maps is clearly homotopic to the identity, for the second we construct the homotopy.

For the points in $\mathcal{R}_\Phi - \mathcal{R}_\Phi \cap_{pair} \mathcal{O}_\Phi$ representative of an equivalence class $[x]_q$ we can define a strong deformation retraction as $F : \mathcal{M}/\mathcal{O}_\Phi \times [0,1] \to \mathcal{M}/\mathcal{O}_\Phi$, where we have $F(x,0) = \mathrm{id}_{\mathcal{M}/\mathcal{O}_\Phi}([x]_p)$ and $F(x,1) = \pi \circ \sigma([x]_p)$.

$$
F([x]_p, t) = \begin{cases} [x]_p \text{ if } x \notin \mathcal{R}_\Phi - \mathcal{R}_\Phi \cap_{pair} \mathcal{O}_\Phi, \\ \gamma_z(t)([x]_p) \text{ if } x \in \mathcal{R}_\Phi - \mathcal{R}_\Phi \cap_{pair} \mathcal{O}_\Phi. \end{cases}
$$

Where $\gamma_z(t)$ is a continuous (in both $t$ and $z$) path from any point $[x]_p \in p \circ q^{-1}([x]_q)$ to the representative $[z]_p$ point. This map starts at $\gamma_z(0)([x]_p) = [x]_p$ and ends at $\gamma_z(1)([x]_p) = [z]_p$ while following a continuous path. Such a path is guaranteed to exists due to the fact that equivalence classes in $\mathcal{R}_\Phi$ arise as a result of a low-rank projection of a convex set, which implies they are convex (and therefore contractible) in $\mathcal{M}/\mathcal{O}_\Phi$ themselves. This proves that there is a homotopy equivalence between $\mathcal{M}/\sim_\Phi$ and $\mathcal{M}/\mathcal{O}_\Phi$. Since homology groups are invariant under homotopy equivalence we get our final result,

$$
H_k(\mathcal{M}/\sim_\Phi) \simeq H_k(\mathcal{M}/\mathcal{O}_\Phi).
$$

$\square$

It is important to state that while we have used convexity as the required property for our proof, a much more general and abstract but also less restrictive property exists. Namely it is necessary that the fibers $\sigma^{-1}([x]_q)$ are contractible. Since $\sigma$ is a surjection and the two quotients are compact metric

spaces, the statement $H_k(\mathcal{M}/\sim_\Phi) \simeq H_k(\mathcal{M}/\mathcal{O}_\Phi)$ follows from an application of the Vietoris-Begle theorem (Spanier (1966)). This fiber condition is always true when the intersections are convex since linear projections of convex sets are convex and their fibers are contractible, but it could also be true when the intersections are not convex.

But how often is it true that $\mathcal{M} \cap G_J$ is convex for any $J$? It turns out that this is always true if $\mathcal{M}$ is convex. Otherwise the probability of this property obtaining likely increases with the size of a network. This is expected to happen since as the number of polyhedra increases, the volume of each polyhedron shrinks and intersects a smaller area of the manifold. This observation is also empirically supported by Figure 9. At this stage this is a heuristic argument and hopefully future research will clarify the extent to which it applies in practically useful neural networks.

Finally, while working directly with the overlap decomposition $\mathcal{O}_\Phi$ is useful for the proofs above, this is not the set that we compute with our algorithm. We actually work with a coarser set that is homotopy equivalent to $\mathcal{O}$ which we define as,

$$\hat{\mathcal{O}}_{\Phi^l} = \{[x] | \Phi(y) \in \bigcap_{i \in I} \Phi(G_{J_i}^l) \text{ for } I \subset \{1, 2, ..., \# \text{ polyhedra}\}$$

$$\text{and } \Phi(y) \notin \bigcap_{i \in K} \Phi(G_{J_i}^l) \text{ for any } I \subsetneq K, |I| > 1\}. \tag{7}$$

This is the same set with the exception that by dropping the condition $\Phi(x) = \Phi(y)$, all points in the maximal intersections of the images of $\Phi$ over different polyhedra $G_{J_i}^l$ are now identified together. Since each equivalence class $[x] = \bigcap_{i \in I} \Phi(G_{J_i}^l)$ is a convex set (intersections of convex sets are convex), by the same argument, there is a homotopy equivalence between $\Phi(\mathcal{M}) \simeq \mathcal{M}/\mathcal{O}_\Phi$ and $\mathcal{M}/\hat{\mathcal{O}}_\Phi$ through contraction of each such intersection to some representative point $[z]$.

**Proposition B.6.** $\mathcal{O}_\Phi$ *is always trivial for single layer networks.*

*Proof.* This is simply a consequence of the fact that overlaps require at least two regions with the same output, but in one layer networks each region is defined by a unique codeword. Therefore, the points in each region are projected to different subspaces and do not overlap. □

## C    EXAMPLES

**Proposition C.1.** *Given a region $L^l_J$ of the local decomposition $\mathcal{L}^l$, there can be two points $x, y \in L^i_J$ for which the maps $\Phi^l|x \neq \Phi^l|y$.*

*Proof.* We can prove this by a simple counterexample. Consider the interval $[-1, 1]$, let's say that there is a piecewise-function,

$$f(x) = \begin{cases} x + 1 \text{ if } x < 0 \\ -x + 1 \text{ if } x > 0. \end{cases}$$

Since this is a piecewise-linear function we know that there is an associated neural network $\Phi : \mathbb{R} \to \mathbb{R}$ with a single output neuron that implements it. We also know that the local decomposition is $\mathcal{L} = L_{\{1\}} = [-1, 1]$. Clearly the two affine functions are not equal to each other. $\qquad\square$

**Proposition C.2.** *The XOR problem, by which we mean finding a map $f : \mathbb{R}^2 \to \mathbb{R}$ such that $f(1, 0) = f(0, 1) > 0$ (we can always rescale if we want this to be equal to 1) and $f(0, 0) = f(1, 1) = 0$, can be made linearly separable by a network with either a trivial overlap or rank decomposition, but not both.*

*Proof.* We will approach this constructively in order to get a better intuition for what is happening in the two types of solutions.

1. Beginning with the pure rank solution, take a single layer ReLU neural network of width two, with weights specified by the matrix,

$$W = \begin{bmatrix} -1 & 1 \\ 1 & -1 \end{bmatrix}$$

and bias $b = -[0.5, 0.5]^T$. As one can see $W[0, 0]^T + b = W[1, 1]^T + b = b$ and since $b$ is negative after application of the ReLU both of these points go to the origin $(0, 0)$. On the other hand, $W[1, 0]^T + b = [-1, 1]^T - [0.5, 0.5]^T = [-1.5, 0.5]^T$ and $W[0, 1]^T + b = [0.5, -1.5]^T$. After application of the ReLU $(1, 0)$ will be on the y axis and $(0, 1)$ will be on the x axis (this is in the space of the neurons in the second layer, see Figure 4 for an illustration). At this point the two classes become linearly separable by the family of hyperplanes $H_c = \{x + y = c | 0 < c < 0.5\}$.

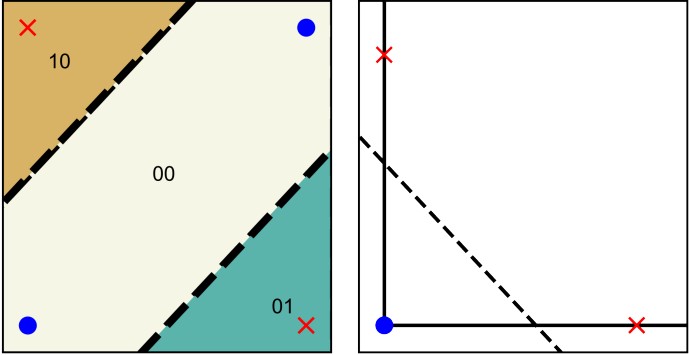

Figure 4: (left) the polyhedral decomposition at the first layer in a pure rank solution of the XOR problem. Numbers indicate the codeword of each region which corresponds to which neurons are active. (right) The linearly-separable representation of the data at the first layer. The interrupted line is one way to separate the data.

Notice that the weight matrix is of rank 1 and the network map $\Phi$ is actually of rank 0 on $(0, 0)$ and $(1, 1)$. Therefore the network splits $\mathbb{R}^2$ into three regions all of which

are low-rank (we take the convention that the dimension of n points is n, since contracting a point to a point does not impact the homology). We can write the rank decomposition $\mathcal{R}_\Phi = \{\{(1,0)\}, \{(0,1)\}, \{(0,0), (1,1)\}\}$ which implies that $H_0(\Phi(XOR)) = H_0(XOR/\mathcal{R}_\Phi) = 3\mathbb{Z}$ as can be found through a relative homology argument.

2. The overlap solution requires more neurons and as we show below - more than one layer. Start with a two layer network with four neurons in layer one and two neurons in layer two. Below we specify the weight matrices and biases of the two networks.

$$
W_1 = \begin{bmatrix} 2 & 1 \\ 1 & 2 \\ -2 & -1 \\ -1 & -2 \end{bmatrix}, b_1 = [-0.5, -0.5, 2.5, 2.5]^T,
$$

$$
W_2 = \begin{bmatrix} 1 & -1 & 0 & 0 \\ 0 & 0 & 1 & -1 \end{bmatrix}, b_2 = [0, 0]^T.
$$

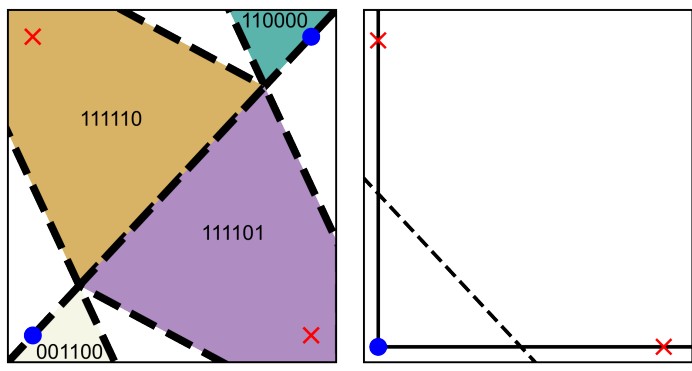

Figure 5: Same as Figure 4 except it shows the polyhedral decomposition at the second layer. Since the blue dots are in different regions they become part of the overlap decomposition.

Applying these maps along with the ReLU's to the four points we get $\Phi(0,1) = (0,1)$, $\Phi(1,0) = (1,0)$ and $\Phi(0,0) = \Phi(1,1) = (0,0)$. Again this is a linearly separable representation of the data that can be separated by the family of hyperplanes $H_c = \{x + y = c | 0 < c < 1\}$. However, the overlap decomposition is now non-trivial $\mathcal{O}_\Phi = \{\{(0,0), (1,1)\}\}$ since every point is within its own region $G_J$ (see Figure 5 for an illustration), yet the blue points end up being glued together. The same homological argument applies.

3. Finally the reason that both decompositions cannot be trivial is a consequence of the fact that we require that $f(0,0) = f(1,1)$ and Theorem B.2 implies that this is the case only when the rank or the overlap decompositions are non-trivial.

$\square$

**Proposition C.3.** *A two layer network with architecture (1,N,1) with $N + \lceil N/2 \rceil = N + M$-linear regions can have up to $\frac{N(N-1)}{2} + \frac{M(M-2)}{2}$-overlap regions.*

*Proof.* With $\sigma =$ ReLU, take the function,

$$
f(x) = \sigma(x) + \sum_{n=1}^{N} (-1)^n \sigma(2x - 2n).
$$

This is a neural network with weights in the first layer $W_1 = [1, 2, 2, 2, ...]$, biases in the first layer $b_1 = [0, 2, 4, 6, ...]$, weights in the second layer $W_2 = [1, -1, 1, -1, ...]$ and no biases in the second

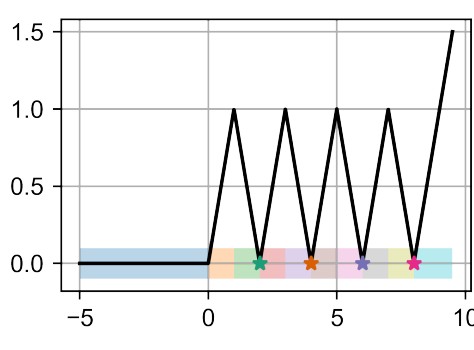

Figure 6: The specified function with $N = 9$, the colored intervals highlight the polyhedral decomposition of this network.

layer. It is directly clear from looking at the graph of this function in Figure 6 that all pairs of the $N$ positive regions overlap, in addition the $\lceil N/2 \rceil$ regions at 0 also overlap. Thus after ignoring symmetric pairs we get the desired upper bound.

□

## D  SIMULATION DETAILS

Here we present all parameters and details for all simulations presented in the results. Where no further explanation is needed, we shall present the parameters in tables.

### D.1  PSEUDOCODE FOR OVERLAP DETECTION ALGORITHM

---
**Algorithm 1** Linear programming for overlap detection

---
  **Input:** Points $X$ from $\mathcal{G}_l$
  **for** $P_i \neq P_j \in \mathcal{G}_l$ **do**
    $\{A_i, b_i\}, \{A_j, b_j\}$ H-representations of $P_i$ and $P_j$
    **for** $\forall y \in P_i$ and $\forall z \in P_j$ **do**
      $C \leftarrow \{A_j x \leq b_j\}$ and $\{\Phi^l_{J_j}(x) = \Phi^l_{J_i}(y)\}$ Add the constraints
      $\text{LP} \leftarrow \min_x(0^T x)|C$ Solve the linear program
      **if** LP is not empty **then**
        $B_y \leftarrow y$ or $B_z \leftarrow z$
      **end if**
    **end for**
    $\mathcal{O} \leftarrow B_y \times B_z$ Add overlapping pairs
  **end for**
  $\mathcal{O} \leftarrow UF(\mathcal{O})$ Union-Find to get overlapping sets
  *Return $\mathcal{O}$*

---

### D.2  NON-LINEAR CURVES SIMULATIONS

We show two example non-linear curves in Figure 1 and ten more in a supplementary Figure 8. The parameters described below apply for each individual example that we present. Since the point of these simulations is to compare persistent homology to our quotient homology based approach, we only use a training dataset. All parameters are specified in the table below.

Table 1: Parameters for non-linear curve simulations

| # SAMPLES | CRITERION | LEARNING RATE | EPOCHS | STOPPING CRITERION | WIDTH | DEPTH | SENSITIVITY |
|---|---|---|---|---|---|---|---|
| 500 | MSE | 1E-4 | 1000 | MSE < 0.00002 | 50 | 3 | 1 |

### D.3  REPRODUCTION OF THE RESULTS OF NAITZAT ET AL.

As mentioned before, we reproduced the results from Naitzat et al. (2020) with fewer samples due to the high computational demand of computing the overlap decomposition. This meant that we also had to reproduce the pre-processing steps in their pipeline. In order to speed up the computation of persistent homology and attain a more robust measure of distance, Naitzat et al. fixed a $k$-nearest neighbors graph at a fixed persistence parameter $\epsilon$ for all datasets. Since the homology groups of the datasets were known a priori, they were able to choose these values on the dataset and carry them forward in the computation of homology groups in deeper layers.

We follow in their steps, but due to the different number of samples we ended up using slightly different parameter values. Below we show the parameter values for which we achieved the ground truth topology (in green), the one we ended up choosing is highlighted in red.

In addition we wanted to avoid issues in homology estimation due to missclassified points, so we excluded all such points from the test set before performing a homology calculation. Given that our networks had almost perfect accuracy, this procedure removed very few points ($7.5 \pm 12.9$). For the calculation of persistent homology we used the **Ripser** package (Tralie et al. (2018); Bauer (2021)).

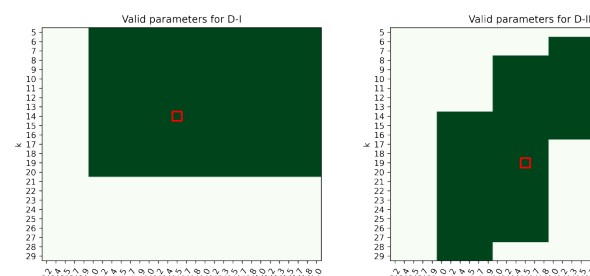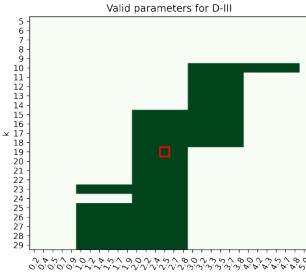

Figure 7: Parameter values that generate the same Betti numbers as the ground truth data. The chosen values were $\epsilon = 2.5$ for all datasets, k=14 for D-I and k=19 for D-II and D-III. These values are highlighted in red.

Table 2: Parameters for Naitzat et al. reproduction results

| DATASET | TRAIN/TEST | CRITERION | LEARNING RATE | EPOCHS | STOPPING CRITERION | # MODELS | WIDTH | DEPTH |
|---------|-----------|-----------|---------------|--------|--------------------|----------|-------|-------|
| D-I | 7800/2200 | CSE | 2E-5 | 5000 | ACCURACY $> 0.999$ | 30 | 15 | 9 |
| D-II | 7500/2500 | CSE | 2E-5 | 5000 | ACCURACY $> 0.999$ | 30 | 15 | 9 |
| D-III | 8000/4000 | CSE | 2E-5 | 5000 | ACCURACY $> 0.999$ | 30 | 15 | 9 |

## D.4 COUNTING REGIONS IN THE SPHERES DATASET

All experiments for spheres of different dimensions $S^1$, $S^2$ and $S^3$ share the same parameters except for the dimension. Also the two classes were balanced, meaning that they contained the same number of samples.

Table 3: Parameters for spheres datasets

| # SAMPLES | CRITERION | LEARNING RATE | EPOCHS | # MODELS | WIDTH | DEPTH | SENSITIVITY |
|-----------|-----------|---------------|--------|----------|-------|-------|-------------|
| 2000 | CSE | 2E-5 | 1000 | 10 | 25 | 4 | 1 |

# E    SUPPLEMENTARY FIGURES

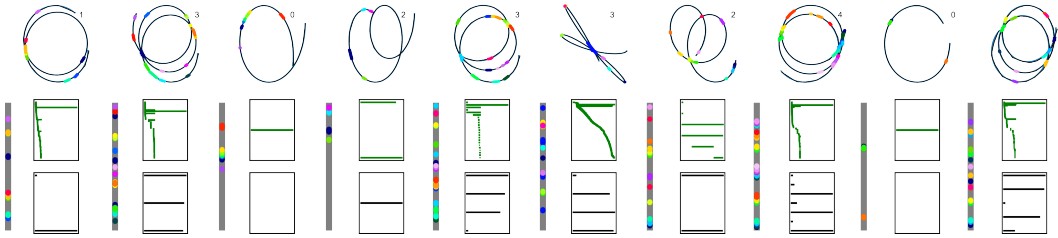

Figure 8: More examples of non-linear manifolds and their homology groups. The numbers in the top right of each manifold show the ground truth number of holes. While our quotient homology approach typically performs better, notice that in the fourth and (arguably) eighth example we see a homological type 2 error.

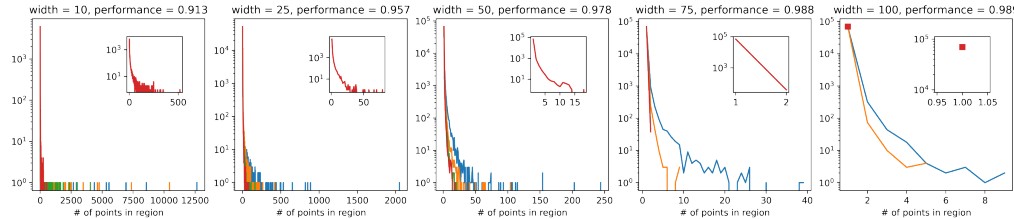

Figure 9: Histograms of the number of points that fall in each polyhedron induced by four layer networks trained and evaluated on all 60000 MNIST digits as a function of width. The colors (blue, orange, green and red) correspond to the respective four layers. One can observe that as the width of the network increases, we see an improvement in performance accompanied by fewer points belonging to each polyhedron. In layers three and four of the widest network, we see that each polyhedron is populated by only one point. This indicates that as networks become wider, the chance of only having convex intersections increases which is consistent with the fact that the volume of the average polyhedron decreases.

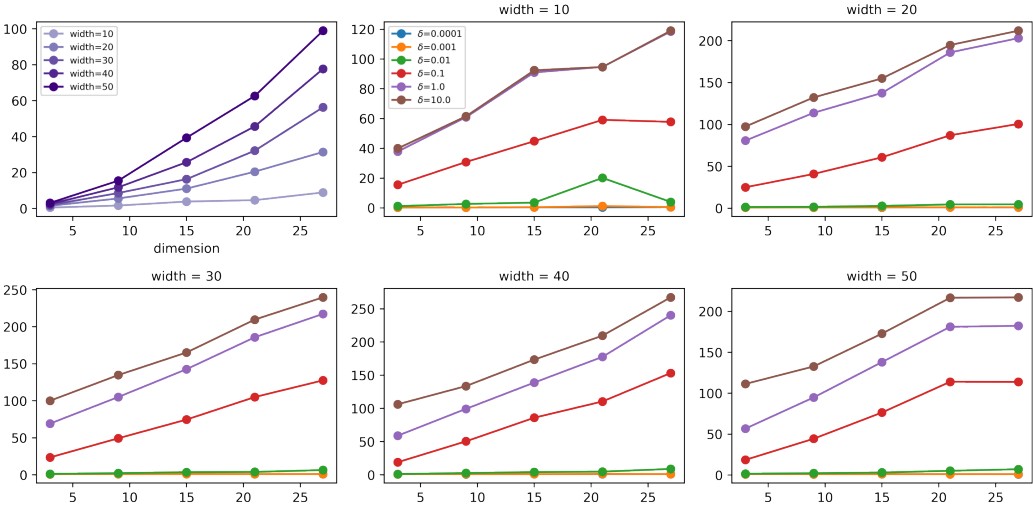

Figure 10: (Top left) Time in seconds to compute the H-representations of all polyhedra in a two layer network as function of width and input dimension on 1000 points sampled from the datasets in section 4.3. The remaining plots show the time it takes to compute the overlap decomposition as a function of width, input dimension and the sensitivity parameter. As can be seen the dependence on width is relatively minor. On the other hand, low values of the sensitivity parameter lead to very fast computing times, at the risk of missing important topological structure. All runs were performed on an Apple M3 Pro CPU with 11 cores.

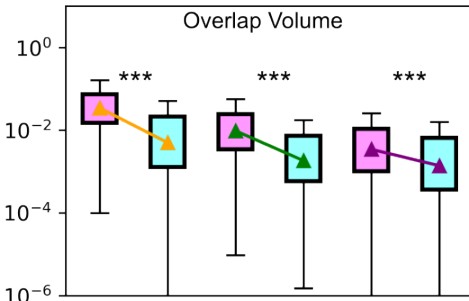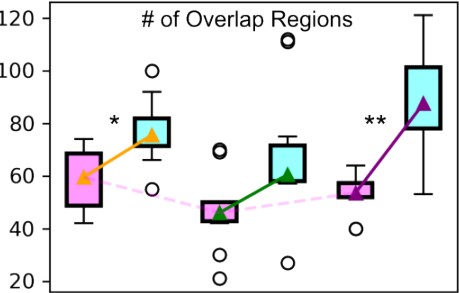

Figure 11: Same plot as in Figure 3 but using an orthogonal initialization (in magenta).

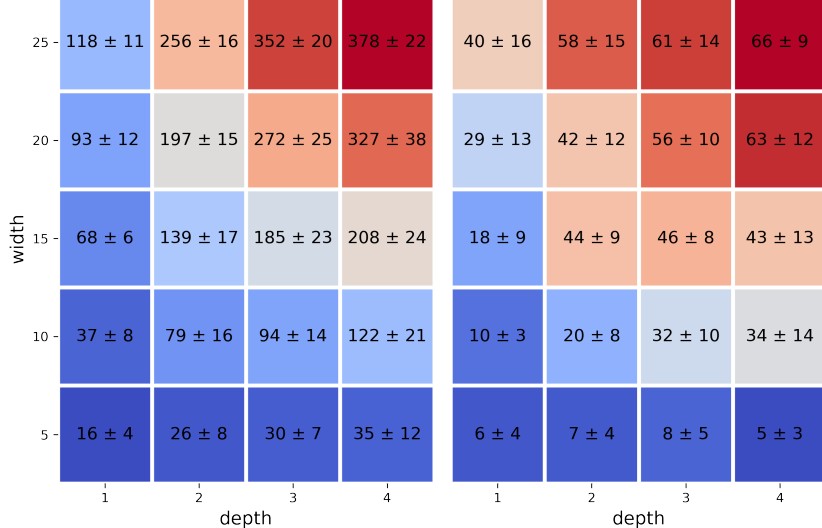

Figure 12: (left) Number of populated polyhedra across 10 randomly initialized networks on the two dimensional data from 4.3 for different widths and depths. Annotations show the mean ± the standard deviations across models. (right) Number of overlap regions in the same models. Compared to the number of populated polyhedra, the number of overlap regions seems to scale non-monotonically.