# OpenReview forum: "A Quotient Homology Theory of Representation in Neural Networks"
_ICLR.cc/2026/Conference — Submitted to ICLR 2026_

### Official Review · Reviewer_pgaX · 2025-10-17

**Soundness:** 3
**Presentation:** 2
**Contribution:** 3
**Rating:** 6
**Confidence:** 3

**Summary:**

The paper proposes a method for calculating the homology of neural representations of data manifolds arising in deep ReLU networks. To this end, it proves that, given a convexity assumption, the homology groups of these representations are isomorphic to the homology groups of the quotient of the data manifold modulo the “overlap polytopal decomposition” induced by the network.

**Strengths:**

- The problem discussed in the paper is relevant due to the importance of understanding the (topology of) neural representations.

- The literature review is remarkably thorough. All the relevant works from the TDA literature are cited and discussed carefully.

- The results in Section 4.2 are particularly intriguing, since it challenges a widespread belief around the change in topology of the data manifold across hidden representations.

**Weaknesses:**

As a main weakness, I find that the practical relevance of the paper might be rather limited, due to its computational burden. As the authors admit, computing the homology via the quotient (modulo the overlap decomposition) is extremely expensive, since it requires computing the polytopes defined by the ReLU network. Even though some techniques are proposed to mitigate the computational cost – e.g., in lines 250-255. Still, the method is so expensive that the authors were not able to fully replicate the results by Naitzat et al. (2020), as discussed in Section 4.2.

Overall, despite the computational burden, I am leaning towards accepting the paper due to its quality, but I am not strongly convinced.

Minor:

- Typo: in lines 157-158, I believe a conjunction is missing to connect the two sentences of the period.

- Typo: in line 161, I believe that $H_k([-\pi, \pi]^2 / \mathcal{O}_\Phi)$ should be removed.

- Typo: on the right-hand side of Equation 5, I believe the superscript $l$ is missing from all the occurrences of $\Phi$.

- Lines 205-2013 talk about H-decompositions, and mention a matrix $A_2$ (Eq. 6). However, these have never appeared before. I understand that they are defined in the appendix (Sec. A.1), but they should be introduced in the main body.

- Typo: on line 320, “Betti” should be capitalized, since it is a proper noun.

- Line 438: the formatting of the exponential breaks the line spacing significantly.

**Questions:**

I do not have specific questions. I would still be grateful if the authors expanded on how expensive is the computation of $H_k(\mathcal{M} / \mathcal{O}_\Phi)$, even just by concrete examples of wall-clock time required, with respect to dataset and network size.

---

> ### Author Response · Authors · 2025-11-18
>
> We wish to thank the reviewer for their time and the encouraging comments. We have rewritten parts of our text, added new intuitive examples (Proposition C.2)  and have corrected all the mentioned typos. Please see the updated submission. Below we address the reviewer’s question.
>
> > I would still be grateful if the authors expanded on how expensive is the computation of , even just by concrete examples of wall-clock time required, with respect to dataset and network size.
>
> We have now extended the discussion with a section about the computational complexity (section 5.2) of our method. We also have wall-clock simulations for networks as a function of width and depth that can be seen in what is now Figure 10.
>
> We hope that extended discussion of computational complexity addresses the reviewers concerns. Please let us know if you have any further questions or suggestions.

---

> > ### Comment · Reviewer_pgaX · 2025-11-24
> >
> > I thank the authors for the additional discussion on computational complexity. I will maintain my (somewhat positive) score.

---

### Official Review · Reviewer_4sh5 · 2025-10-31

**Soundness:** 2
**Presentation:** 2
**Contribution:** 2
**Rating:** 4
**Confidence:** 2

**Summary:**

Deep neural networks are hard to understand theoretically. One such way to study them in a more principled manner is using Topological Data Analysis (TDA). Specifically for deep learning, it has been used to study training trajectories and weights. It's typically based on a concept known as Persident Homology. Despite its promise, it is known to have some issues (e.g., sensitivity to outliers, and contamination from geometric features).

This paper proposes an alternative to the methods based on persistent homology. In particular, it focuses on ReLU NNs, and how do they change the topology of data. Crucially, the method doesn't use distances. ReLU NNs split the input space into many convex regions (polyhedra); when many of these regions map to the same output, the network is "gluing" inputs. The paper defines the set of these gluings as the overlap decomposition.  The authors propose detecting these overlaps with linear programming, and show empirical results on toy data, where the methods seem to avoid geometric artifacts (as opposed to persistent homology methods). They suggest that the "topological simplification" happens more gradually than what prior work suggested.

**Strengths:**

- Creative proposal, moving beyond persistent homology.
- Empirical results accompanying the theoretical ones. Examples where persistence may report loops due to geometry, while the quotient view does not.
- Interestingly, uses piecewise-linear / polyhedral view of ReLU nets to define overlaps.
- Training dynamics findings: Suggests slower topological collapse across layers than earlier results, this could be relevant in practice.
- Generally well-written.

**Weaknesses:**

- Strong assumptions: Convexity is a strong assumption, will it work for natural data?
- Scalability: I also formulated it as a question below, because I'm not certain, but the method does not seem to be particularly scalable (LP method, pairwise).
- Only toy data. It wouldn't particularly worry me if not for the strong assumptions.

- 5.2 TOPOLOGICAL VERSUS GEOMETRIC TRANSFORMATIONS" is very interesting. Still, I'm not convinced why in practice reason geometric contamination would be an issue


"An intuitive illustration of this theorem can be seen in panel A of Figure 1. One might object
that this convexity condition does not usually obtain. However, we note that given the highly
overparameterized nature of neural networks and the increasing number of polyhedra within a given
volume (Hanin and Rolnick (2019)) this condition usually obtains. It also turned out to be empirically
true in the simulations performed in this paper and the additional convexity analysis found in Figure 6
of the Appendix. Furthermore, if we are working with a convex input manifold, this condition always
obtains. As seen in Beshkov and Einevoll (2024) and in our simulations the cases in which this
condition will likely not be satisfied is if one is working with highly non-convex, high-dimensional
input domains and insufficiently wide networks." this paragraph is quite crucial and not very easy to follow

Note:The paper offloads a significant amount of content to the Appendix. There are crucial details there about proofs; I haven't had the cycles to take a deep look at the appendices.

Misc.:
- “Betti numbres” -> “numbers”
- “botton right” -> “bottom right”

**Questions:**

Does this method scale? What are the crucial elements to scale it?

What would be the main challenges for extending this method to non-ReLU NNs?

FIgure 3, despite the caption, is quite hard to follow. Is there any alternative result visualization that would be easier to follow? Or could you provide a more intuitive explanation?


"the cases in which this
condition will likely not be satisfied is if one is working with highly non-convex, high-dimensional
input domains " what would be some examples of domains with these properties? Isn't this how some real-world data would present?

Misc.:
- “Betti numbres” -> “numbers”
- “botton right” -> “bottom right”

---

> ### Author Response · Authors · 2025-11-18
>
> We wish to thank the reviewer for dedicating their time and attention to our paper. Below we do our best at addressing their concerns.
>
> > Does this method scale? What are the crucial elements to scale it?
>
> We have now added section 5.2 to the Discussion, where we describe how our algorithm scales with network and data size. We also propose two ideas that would help improve its scaling as described in that section.
>
> > What would be the main challenges for extending this method to non-ReLU NNs?
>
> Given an activation that is defined piecewise and is non-decreasing (or non-increasing), the input space can always be piecewise decomposed. In the non-linear case the notion of rank is not clearly defined and the rank decomposition has to be defined differently. Moreover, if the function is non-linear, linear programming methods become less accessible and it is not obvious how to algorithmically identify overlaps.
> For activations such as Sigmoid or TanH, there are no overlaps. This is due to the fact that they are injective functions and thus do not change the topology of the input manifold, except through low-rank weight matrices (we mean the actual weight matrices, not the local maps we discuss in the rank decomposition). Other activation functions such as GELU are not injective and would require their own treatment.
>
> > FIgure 3, despite the caption, is quite hard to follow. Is there any alternative result visualization that would be easier to follow? Or could you provide a more intuitive explanation?
>
> We agree that there is a lot happening in the figure and have modified the figure and the caption to be a bit more clear. Here we provide an additional explanation. We study the overlap decomposition before and after training on datasets of inscribed spheres of different dimension as described in section 4.3. In the top row we show the polyhedral regions in which the underlying space is split into on one example run. The magenta plots correspond to the pre-training decomposition and the cyan ones to the post-training one. What we see from this example is that the regions shrink after training. This is then quantified in the plots below, by looking at the volume and number of overlaps for all performed simulations. We have tried to clarify the figure, but please let us know if it is still difficult to interpret.
>
> >  what would be some examples of domains with these properties? Isn't this how some real-world data would
> present?
>
> First off, we wish to stress that our assumption is much weaker than the convexity of the whole data. What we require is that each polyhedron in $\mathbb{R}^{n_0}$ intersects the data in a convex manner. We provide empirical support of the convexity assumption in MNIST for different networks in Figure 9. Here we also add a heuristic argument for why this assumption will quite often apply in real data.
>
> According to Hanin and Rolnick (2019), the expected number of polyhedra generated by a network is on the order of $N = (M^{n_0})/{n_0}!$, where M is the number of neurons and $n_0$ is the input dimension. If we assume that the data is in a hypercube of volume $b^{n_0}$, then the average number of polyhedra in the hypercube will be proportional to $\frac{b^{n_0}}{N}$. Increasing the number of neurons polynomially shrinks the volume of the intersection between the data and these polyhedra. So if B is a 3d cube and we have a network with 1000 neurons, then the average region will have a volume on the order of $10^{-6}$. The data would need to have noticeable curvature at this scale for our assumption to fail.
>
> The worst case example for our method is fractal data, in that case the assumption is always false. We can even show that given enough neurons and non-fractal data, there is always a network that covers the data in a way that is consistent with the assumption. As long as we can cover the data with arbitrarily small convex sets we can ensure that they cover the data in a convex way. There exist piecewise functions over that covering and therefore there also exists a network realizing them (every piecewise function can be realized by a network due to Arora et al. (2018)).
>
> We have rewritten parts of the text to improve readability and better motivate the implications of our work. We have also added examples (see Proposition C.2) that are useful for building intuition about the rank and overlap decomposition. Please see the updated submission. We hope that we have clarified that the convexity assumption is not as strong as one might expect. Furthermore, the section on scalability provides a path for scaling our method to more complex settings. Please let us know if you have any further questions or suggestions
>
> References:
>
> Arora, R., Basu, A., Mianjy, P., and Mukherjee, A. (2018). Understanding deep neural networks with rectified linear units.
> Hanin, B. and Rolnick, D. (2019). Deep relu networks have surprisingly few activation patterns. Advances in neural information processing systems, 32.

---

### Official Review · Reviewer_mWrc · 2025-10-31

**Soundness:** 2
**Presentation:** 2
**Contribution:** 3
**Rating:** 4
**Confidence:** 4

**Summary:**

This paper proposes an alternative homology theory, quotient homology, to study representation spaces of neural networks over the existing standard homology theory (persistence) for computational settings, which has recently become popular, especially in machine learning.  Quotient homology has some desirable properties, such as improved robustness.  Aspects of the resulting measure, termed overlap decomposition, are studied for trained versus untrained networks with some discernible patterns.

**Strengths:**

- A good attempt was made to make some quite intricate and theoretical concepts accessible.

**Weaknesses:**

- At times the language is rather awkward, e.g., line 276: "One might object that this convexity condition does not usually obtain." and further on in the same paragraph.  Also, the tone and presentation varies from quite formal to almost conversational, which is quite distracting.  I would consider the writing to be a major and quite serious weakness.
- Possibly an artifact of the awkward writing, but aside from mathematical interest, the intuition for studying such mathematical quantities and what it means for the learning process is quite unclear.  Although the mathematics is indeed interesting, and some elegant concepts are adapted (and hence my higher contribution score), the implication of the results and these concepts are not clearly interpretable to a machine learning audience such as ICLR (and hence my lower soundness score).

**Questions:**

- Why is "informal" specified after Theorem 3.1?  What does this mean?
- Can something be said about the differences in computational expense of persistent homology (known to be very expensive) and the proposed quotient homology approach?  Especially in the case of high dimensional input regions, when persistent homology can be very computationally intensive?

---

> ### Author Response · Authors · 2025-11-18
>
> We wish to thank the reviewer for their time and effort in reviewing our manuscript. We apologize that the writing felt awkward. Our goal has been to balance rigor with accessibility for the general machine learning audience, but we can see that the more “conversational” writing can be distracting. We have now gone through the whole text and have done a major rewrite where we have removed more conversational phrasing such as “One might object” or “This is already encouraging”. Please see the newly uploaded version. Below we address the reviewer's questions.
>
> > Why is "informal" specified after Theorem 3.1? What does this mean?
>
> We mean that this is an intuitive explanation of the theorem rather than a rigorous mathematical statement which is stated in Theorem B.2. in the Appendix. We have now added a sentence before the theorem in the main text pointing out that it is there to build intuition.
>
> > Can something be said about the differences in computational expense of persistent homology (known to be very expensive) and the proposed quotient homology approach? Especially in the case of high dimensional input regions, when persistent homology can be very computationally intensive?
>
> Given that the question of computational complexity has been raised by several reviewers we now address this question in section 5.2 of the discussion. It is hard to compare the complexity of our method to persistent homology as that depends only on the data and its representation in latent space, whereas our algorithm also incorporates the network architecture.
>
> Besides rewriting the text in a less conversational tone, we have also added: a new section on computational complexity, several intuitive examples showing how the rank and overlap decompositions lead to different solutions of the XOR problem (Proposition C.2. on pages 22 and 23) and an exploration of expressivity in terms of linear regions and its relation to expressivity in terms of overlap regions (section 5.3). We believe that the improved readability and these new additions help the machine learning audience gain a more clear understanding of why the objects we propose are relevant and worth studying. We hope you agree, please let us know if you have further questions or suggestions.

---

### Official Review · Reviewer_qz89 · 2025-11-14

**Soundness:** 3
**Presentation:** 2
**Contribution:** 3
**Rating:** 6
**Confidence:** 3

**Summary:**

This paper introduces a novel theoretical framework connecting ReLU neural networks with quotient homology theory. Building on the fact that ReLU networks correspond to continuous piecewise-linear maps, the authors define an equivalence relation on the input manifold based on regions where the network is non-injective. They propose the overlap decomposition capturing intersections between affine regions, and prove that the homology of neural representations is isomorphic to the quotient homology.
They develop a computational method to identify overlaps and experimentally demonstrate the framework on toy datasets. Their method is shown to isolate topological features while removing geometric artifacts. The paper further analyzes how the overlap decomposition evolves during network training and how it correlates with expressivity and non-injectivity.

**Strengths:**

The paper is original in proposing quotient homology as a new framework for analyzing neural representations and supports this with rigorous definitions and theorems that connect non-injectivity to topological change. It offers clear conceptual distinctions between geometric and topological effects, reinforced by insightful visualizations that make these differences intuitive. Overall, the approach has strong potential to influence future work on expressivity and manifold transformations in deep networks.

**Weaknesses:**

1. The paper’s main theorem depends on the assumption that each intersection $M\cap G_{j}$ is convex, but this condition is unlikely to hold for realistic, high-dimensional, and non-convex input domains. The authors provide only low-dimensional toy examples to justify the assumption, leaving unclear how broadly the theoretical result applies in practice.
2. The theory identifies two distinct mechanisms of topological change—rank decomposition and overlap decomposition—but the experiments only analyze the overlap component. As a result, the empirical section does not isolate how much of the observed homology is attributable to rank drops versus overlaps. There are no controlled settings where one source is varied while the other is held fixed, nor measurements of how often rank changes occur in practice. This makes it difficult to assess the relative importance of each mechanism or to validate the full theoretical decomposition experimentally.
3. Grammar errors:
Line 070: which except for a few works -> which, except for a few works
Line 255-256: choosing a different δ’s -> choosing different δ’s
Line 297-298: a MSE loss -> an MSE loss
Line 306-308: both persistent homology as well as quotient homology -> both persistent homology and quotient homology
Line 341: botton right -> bottom right
Line 348-349: Betti numbres -> Betti numbers
Line 377: the number of overlap regions are -> the number of overlap regions is

**Questions:**

1. The convexity assumption plays a central role in both the overlap-detection algorithm and the isomorphism theorem, but it seems stronger than necessary from a topological standpoint. In particular, The convexity condition in the central theorem (Theorem 3.2) can be relaxed to contractibility as in Theorem B.5. Can the convexity requirement $M\cap G_{J}$ be relaxed, e.g., to contractibility or another weaker “cell-like” condition, and if not, could the authors clarify precisely where convexity is essential in the proof and why weaker assumptions would fail?
2. The theoretical framework distinguishes between rank-induced and overlap-induced topological changes, but the experiments focus only on overlaps. Could the authors provide ablations or analyses that isolate the role of rank decomposition—e.g., settings where rank varies while overlaps are minimal—and clarify how often rank drops occur in practice and how much they contribute to the resulting homology? Or if it is not possible, could the authors provide appropriate reasons?
3. How does the computational complexity of the overlap decomposition scale with dimension and layer count?
4. Is there a relationship between overlap counts and expressivity measures like linear region counts or VC dimension?
5. How sensitive is the quotient homology computation to small perturbations in weights or inputs? Do the authors have any insights on this?

---

> ### Author Response · Authors · 2025-11-18
>
> We wish to thank the reviewer for their helpful suggestions and thorough reading of our work. We have tried to implement their comments and believe our paper has noticeably improved as a result.
>
>
> **Q1.** Thank you for pointing this out, the statement of Theorem 3.2 is the correct one and leaving “(or more generally - contractible)” in Theorem B.5. was an oversight by us. We never use contractibility of the intersections in the proof, but rather require that the fibers $ \sigma^{-1}([x]_q) $ are contractible. If this is not the case, then there is no way to construct a path between any $[x]_p$ and the representative point $\pi([x]_q)=[z]_p$ and therefore $\pi \circ \sigma$ will not be homotopic to the identity which means that the rank decomposition will matter. This is different from the intersections being contractible, as our oversight might have implied. For an example, think of a 3d spiral being projected down to a circle, the fibers will be disconnected points on the spiral even though the spiral itself is contractible.
> Convexity guarantees that the fibers are contractible, since the fibers of low-rank maps on convex sets are convex themselves. To be explicit, the convexity is used in lines 1070-1072 (in the newest version) to justify the construction above it.
>
>
> The requirement that $\sigma^{-1}([x]_q)$ is a more abstract property which is much weaker than convexity.  The downside is that it is much less intuitive and it is even less clear what it means in practice. To make this clear we have now clarified the proof and have added an alternative explanation in terms of this more abstract property in the paragraph after the proof.
>
>
> **Q2.** It is true that we have mainly focused on the overlap decomposition as its impact is easier to track and under the convexity assumption the rank source can be ignored. We have isolated the role of the rank decomposition in the empirical results shown in Figure 2, but we agree that a further explanation and intuition for these two sources will significantly improve our paper. Therefore, we have now added two intuition-building examples (Proposition C.2. on pages 22 and 23) of networks that solve the XOR problem using a rank and an overlap strategy. Furthermore we have tried to do a better job at discussing how these two sources arise and interact throughout the text.
>
>
> **Q3.** This question has been raised by several reviewers and we have now added its own section (5.2) in the discussion.
>
>
> **Q4.** There definitely is some relationship between linear and overlap regions and we think this is a fascinating direction for future work since it would track the expressivity for non-injective maps in particular. The exact relationship is probably quite intricate, but we have now added some simple relations between the two. For example in Proposition B.6 on page 21 we show that in single layer networks with architecture $(n_{in},N)$, there are no overlapping regions so such networks are less expressive for non-injective functions (the rank source can still make them non-injective). For networks of architectures $(1,N,1)$ we can establish a constructive lower bound on the overlapping regions as we show in Proposition C.3. on page 23. The relation to the VC dimension is less intuitive to us, but we hope to understand that in future work.
>
>
> **Q5.** At this stage we do not have rigorous arguments for this, but it should be robust as long as the inputs remain in the same polyhedra and the weights maintain the arrangement of the polyhedra. For arrangements in general position, which are dense in the space of neural networks (see Grigsby and Lindsey (2022)), we expect that robustness will hold, but there might be critical arrangements where things change. This is another avenue in which future work is needed.
>
>
> We hope that our response has answered your questions. We certainly believe that the new additions in terms of text, intuitive examples and proofs have made our manuscript much better. Please let us know if you have any further questions or suggestions.
>
>
> References:
>
> Grigsby, J. E. and Lindsey, K. (2022). On transversality of bent hyperplane arrangements and the topological expressiveness of relu neural networks. SIAM Journal on Applied Algebra and Geometry, 6(2):216–242.

---

### Meta-Review · Area_Chair_bG9y · 2026-01-07

**Summary:**

The article studies the homology groups of neural representations and proposes a method to compute Betti numbers intrinsically based on an overlap decomposition.

Reviewers indicate concerns with the assumptions for the theoretical analysis, scalability, and interpretation of the results in a machine learning context.

The article advances interesting ideas that could turn out to be valuable. However, I believe the reviews point at valid limitations, particularly in regard to computational complexity, and that addressing them as well as the practical implications more thoroughly would strengthen the contribution. Therefore I am recommending reject with an encouragement to revise and resubmit.

**Reviewer Concerns:**

Reviewer 4sh5:
Does this method scale? What are the crucial elements to scale it?
-> added section 5.2 to the Discussion, where we describe how our algorithm scales with network and data size.

Reviewer pgaX:
practical relevance of the paper might be rather limited, due to its computational burden
-> extended the discussion with a section about the computational complexity

**Reviewer Scores:**

For each review, specify how you think the reviewer would have changed their score if they had been able to participate fully in the discussion.

Reviewer qz89: 6 -> 6
Reviewer mWrc: 4 -> 4
Reviewer 4sh5: 4 -> 4
Reviewer pgaX: 6 -> 6

---

### Decision · Program_Chairs · 2026-01-26

Reject